# Oil palm expansion and deforestation in Southwest Cameroon associated with proliferation of informal mills

Elsa M. Ordway [1,2], Rosamond L. Naylor[1,3,4], Raymond N. Nkongho[4,5] & Eric F. Lambin[1,3,6]

Oil palm expansion resulted in 2 million hectares (Mha) of forest loss globally in 2000–2010. Despite accounting for 24% (4.5 Mha) of the world's total oil palm cultivated area, expansion dynamics in sub-Saharan Africa have been overlooked. We show that in Southwest Cameroon, a top producing region of Africa, 67% of oil palm expansion from 2000–2015 occurred at the expense of forest. Contrary to the publicized narrative of industrial-scale expansion, most oil palm expansion and associated deforestation is occurring outside large agro-industrial concessions. Expansion and deforestation carried out by non-industrial producers is occurring near low-efficiency informal mills, unconstrained by the location of high-efficiency company-owned mills. These results highlight the key role of a booming informal economic sector in driving rapid land use change. High per capita consumption and rising palm oil demands in sub-Saharan Africa spotlight the need to consider informal economies when identifying regionally relevant sustainability pathways.

[1] Department of Earth System Science, Stanford University, 473 Via Ortega, Stanford, CA 94305, United States. [2] Department of Organismic and Evolutionary Biology, Harvard University, 26 Oxford Street, Cambridge, MA 02138, United States. [3] Stanford Woods Institute for the Environment, Stanford University, 473 Via Ortega, Stanford, CA 94305, United States. [4] Center on Food Security and the Environment, Stanford University, 616 Serra Street C100, Stanford, CA 94305, United States. [5] Department of Agronomic and Applied Molecular Sciences, University of Buea, Buea, Cameroon. [6] Georges Lemaître Earth and Climate Research Centre, Earth & Life Institute, Université catholique de Louvain, Place L. Pasteur 3, Louvain-la-Neuve 1348, Belgium. Correspondence and requests for materials should be addressed to E.M.O. (email: elsa.ordway@gmail.com)

Oil crops are the leading cause of global land use change. Between 1970 and 2016, oil crops expanded by 150 million hectares (Mha); resulting from a 280% increase in soybean production and 220% increase in palm oil production from 1990 to 2010[1]. Of the four most rapidly expanding oil crops (soybeans, rapeseed, sunflower, and oil palm), oil palm (*Elaeis guineensis*) is the second most significant driver of deforestation, accounting for 2 Mha of forest cleared in 2000–2010 and 1.77 GtCO$_2$e of global greenhouse gas emissions during the same period[2,3]. Growing international attention focused on industrial-scale expansion led several corporations to commit to reducing or eliminating deforestation associated with palm oil production from their supply chains[4,5]. Because most rapid deforestation driven by oil palm growth has been concentrated in Indonesia and Malaysia, recent expansion in sub-Saharan Africa, the geographic origin of the crop, has largely been overlooked.

Although Africa's contribution to global palm oil supplies atrophied from 77% in 1961 to 4% in 2014, the continent still holds 24% (4.5 Mha) of the world's total oil palm cultivated area across climatically suitable regions of West and Central Africa[6]. Expansion accelerated in several countries from 2000–2014, led by Cameroon[7], coinciding with an increase in global palm oil trade due to increased profitability[1,8]. Palm oil has long been consumed throughout West and Central Africa. By 2050, edible oil consumption across the continent is projected to triple relative to 2013[1]. This is due in part to high per capita consumption, population growth projections, and increases in per capita GDP. Areas of oil palm suitability in Africa overlap with Congo Basin tropical rainforests, which account for an estimated 25–30% of the world's tropical forest carbon stocks[9–11]. Whether new oil palm plantations in the region are predominantly replacing forests or other crops on previously cleared lands is poorly understood.

Despite a lack of information on current oil palm expansion in Africa, 10 countries are proactively seeking strategies for sustainable development of their oil palm sectors through the Africa Palm Oil Initiative[12]. These efforts draw on international standards established in response to extensive forest conversion in Indonesia and Malaysia, with the most notable examples being the Roundtable on Sustainable Palm Oil (RSPO) and the High Conservation Value (HCV) and High Carbon Stock (HCS) approaches[13–15]. Sustainability strategies to date have focused on industrial-scale oil palm expansion[16–20], although research increasingly shows smallholders are an engine of growth[1,21–23].

Because milling constitutes a critical step in palm oil production, sustainable sourcing practices are targeting industrial mills as a strategic point for intervention. Upon harvest, oil palm fresh fruit bunches (FFB) rapidly decay, degrading the quality of the palm oil produced[24]. To prevent post-harvest losses, fruit is typically delivered to a mill within 48 h. Producers are significantly constrained by the location of milling facilities due to this time limit. Expansion pathways thus require consideration of both the producer and mill.

Economies of scale suggests that internal production or processing costs decline as the scale or output of a firm increases. Here, firm refers to an individual or company in the context of production, or a mill in the context of processing. Theoretically, large, well-organized agricultural companies are more likely to expand given their competitive advantage over individual producers cultivating at smaller scales. The prevalence of industrial-scale expansion has been demonstrated empirically in Southeast Asia and parts of South America[17,21].

Processing-related internal economies of scale imply that most expansion would occur near high-efficiency mills, given their greater milling capacity. Agglomeration economies, which arise from industry-level benefits that occur when different actors within an industry operate in close proximity, also supports the expectation that clustering is likely near high-efficiency mills. Silicon Valley is a widely cited example of this phenomenon[25,26], while the clustering of the soybean industry in Mato Grosso, Brazil is a well-known agricultural case[27]. This density of industry is largely attributable to transportation cost savings and benefits from shared suppliers, labor, and knowledge[28,29].

In the 1970s, large, agro-industrial companies cultivated 90% of the oil palm in Cameroon[30]. Following a government-led rural financing program from 1978 to 1991, the area cultivated by non-industrial producers increased by 570%, now accounting for an estimated 70% of the oil palm planted in Cameroon[31,32]. The financing program, known as FONADER (National Fund for Rural Development), expanded non-industrial oil palm plantations by providing credit, inputs, and technical support through relationships with companies[33]. To address a national vegetable oil deficit, efforts to support small-scale producers were renewed in 2003[33,34]. Although oil palm cultivation in Cameroon is carried out across a range of production scales[35], recent foreign investment in Southwest Cameroon by a third agro-industrial company (Sithe Global Sustainable Oils Cameroon, Ltd.) has led to concern of industrial-scale expansion[36].

Before government interventions, two companies (PAMOL and Cameroon Development Corporation) owned and operated nearly all palm oil mills in Southwest Cameroon. As the number of producers grew, niche milling opportunities emerged from inefficiencies including delayed payments to farmers, high transportation costs, and an inability of existing mills to process peak season production[37,38]. Unlike industrial mills, which are more comparable to mills operating in Indonesia and Malaysia, most recently established non-industrial mills are unregulated and vary widely in scale and the quality of oil produced, ranging from completely manual to fully mechanized systems.

Here, we explore whether recent patterns of oil palm expansion in Cameroon, Africa's third largest palm oil producing country, conform to the widely publicized view that large companies drive most expansion and associated deforestation. We examine expansion in the context of producer decision-making from three angles: (1) internal economies of scale related to production costs at the farm level, (2) internal economies of scale related to milling and processing costs, and (3) agglomeration economies (i.e., costs and benefits from external, industry-wide economies of scale). This study presents a spatially explicit analysis of recent oil palm expansion in Southwest Cameroon, a major production region in the Congo Basin. Using Landsat satellite imagery, we measure oil palm expansion and associated deforestation inside and outside agro-industrial concessions from 2000–2015 and analyze the relationship between different milling systems and expansion.

## Results

**Area expansion of oil palm.** Oil palm cultivation is concentrated in the Southwest, Littoral, and Central Regions of Cameroon[39]. Expansion began accelerating around 2005 in Cameroon in parallel with production and consumption growth, while on-farm yields stagnated and declined (Fig. 1). Using Landsat satellite imagery, we mapped expansion in the Southwest Region from 2000–2015 (Fig. 2), which encompasses 40% of national oil palm production[40] and was 86% forested in 2000. Roughly 81% of the region is suitable for oil palm cultivation[41]. To create a spatially explicit dataset of palm oil processing facilities, we mapped all mills in the region, totaling 498. The two more established agro-industries operating in Southwest Cameroon own five mills in the region (Supplementary Fig. 1). The third agro-industry, more recently established in the region, had not built a mill by the end of the study period in 2015. These five industrial mills were the

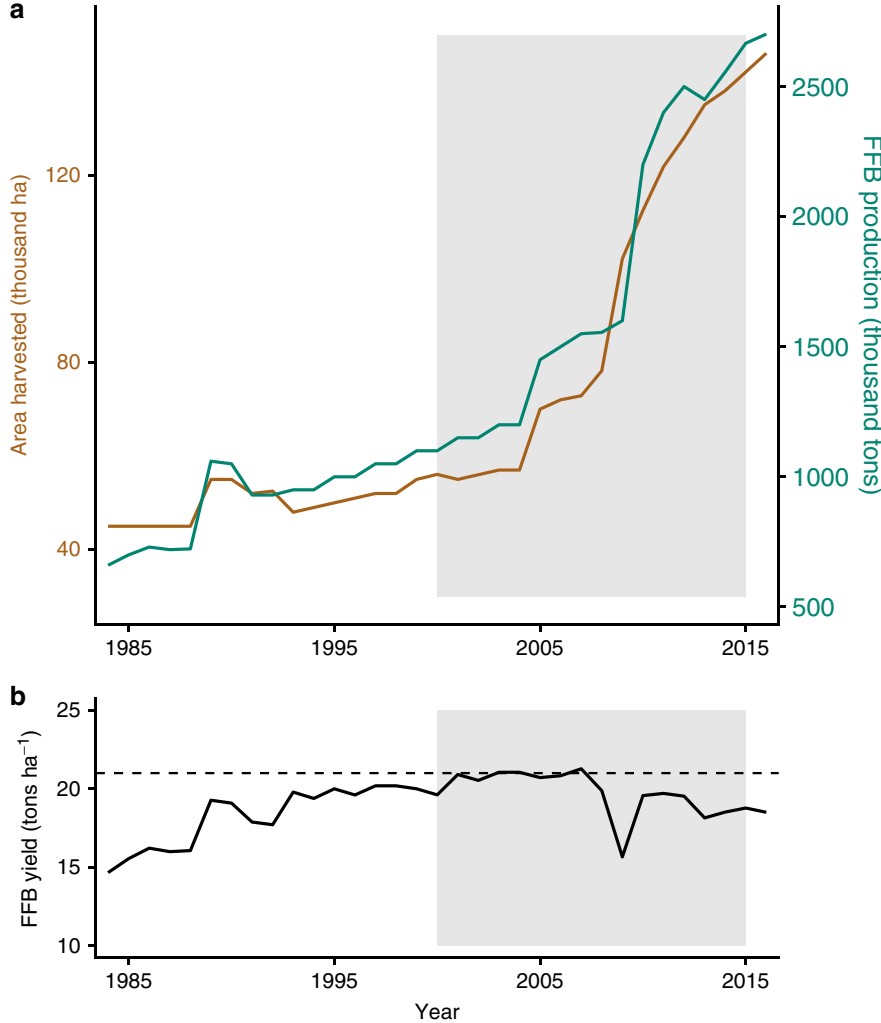

**Fig. 1** Oil palm area expansion (extensification) in Cameroon. **a** Oil palm production (green) and area harvested (orange) since 1985 for the country of Cameroon, with the time-period considered in this study (2000–2015) highlighted in gray[6]. **b** On-farm fresh fruit bunch (FFB) yield in Cameroon since 1985. For comparison, the horizontal dotted line represents the FFB yield in Indonesia[54]. Note: yield values from FAOSTAT[6] reported in **b** should be viewed with caution, as it is difficult to obtain quality data on yield, especially from smallholder farmers. FAOSTAT yield values are much higher than yields observed in the field[30,35]

highest efficiency mills in terms of capacity and extraction rate and contributed to roughly one-third of the region's total oil palm production. We thus refer to these as high-efficiency industrial mills, and the remaining 493 mills as low-efficiency informal mills due to the lack of regulation and informal economy that is associated with them. We mapped oil palm expansion across the entire region and deforestation in areas where expansion occurred. Using these maps, we examined relationships between different types of mills and oil palm expansion, relative to other factors we expect to be associated with expanding cultivation, including market access, indicators of agroecological suitability, and the location of concession areas leased by agro-industries (Supplementary Table 3). In addition to mill type, the oil extraction rate of the mill closest to each grid cell was used as a proxy for examining the influence of milling efficiency on the location of expansion and deforestation. A detailed description of all variables and analyses is included in the Methods section.

**Deforestation and informal mill proliferation.** Our results show that, since 2000, two-thirds of oil palm expansion in Southwest Cameroon occurred at the expense of forest (Fig. 3). Nearly twice

as much land converted to oil palm came from forests relative to the conversion of other land cover types (i.e., other agricultural areas that had previously been cleared). This figure holds true both inside and outside industrial concessions. Contrary to what one would expect from production-related internal economies of scale, however, over two-thirds of all expansion occurred outside industrial concessions. This expansion coincided with a boom in new mills, almost all of which were informal, low-efficiency, and non-industrial facilities (Supplementary Fig. 2). Of the 53% of mills that reported their year of establishment, 95% were built in the year 2000 or later. At full capacity, the five industrial mills combined can accommodate 155 tons FFB hour$^{-1}$. Four of these industrial mills were established prior to 2000, operating at a combined capacity of 140 tons FFB hour$^{-1}$. The 493 non-industrial mills, spread across the entire region, can accommodate a combined capacity of 398 tons FFB hour$^{-1}$. Roughly 97% of the 262 informal mills with a recorded year of establishment were installed post 2000, indicating that most increased milling capacity since 2000 stems from an increase in distributed processing at hundreds of low-efficiency mills.

Results from the spatial analyses indicate a large, significant relationship between informal mills and recent expansion and

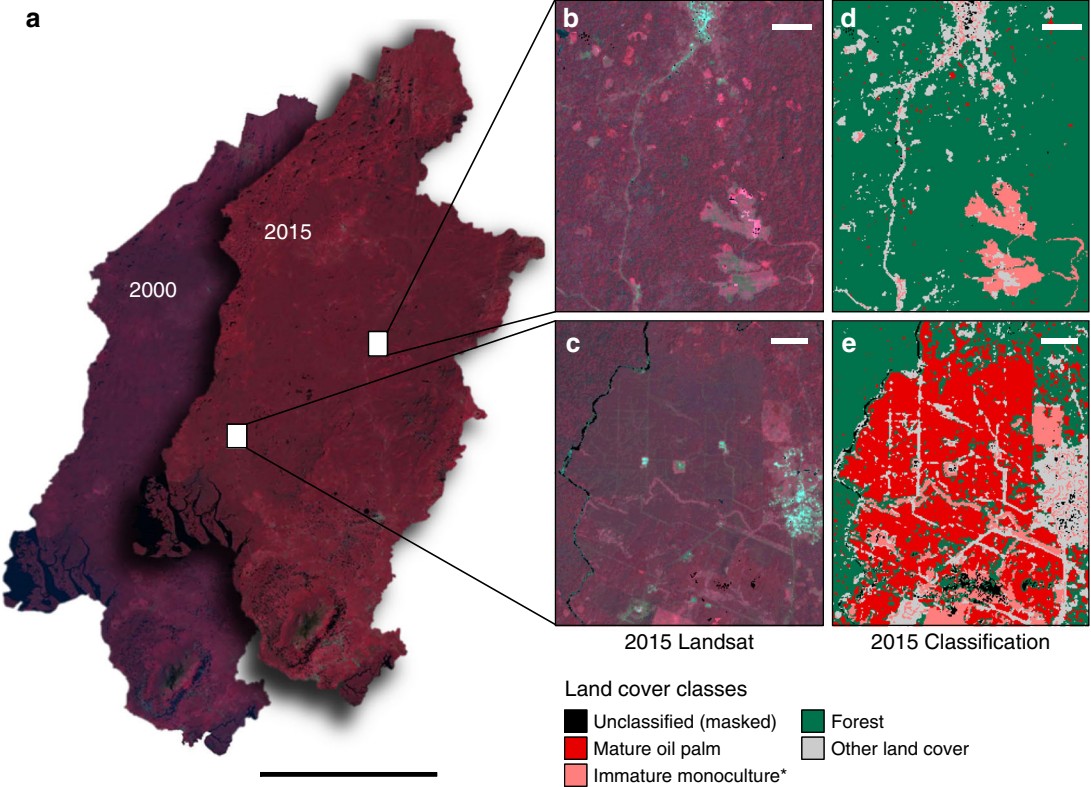

**Fig. 2** Oil palm detection using remote sensing. **a** Pixel composites of Landsat imagery for the years 2000 and 2015 used to classify oil palm in Southwest Cameroon. Scale bar: 80 km. **b–e** Two locations with known oil palm plantations in 2015 illustrated in the Landsat 8 pixel composite (**b**, **c**) and the random forest classification output (**d**, **e**). *We estimated that 65% of immature monoculture pixels were immature oil palm. Scale bars: 1 km

deforestation. Findings highlight not only conversion near informal mills, but also a significant influence of mill type. We focus on variables that were statistically significant at Wald test, $p < 0.05$, with odds ratios and 95% CIs that do not overlap with 1 (Fig. 4, Supplementary Table 1). Odds ratios highlight the influence of each variable on the odds of oil palm expansion (Fig. 4a) or deforestation (Fig. 4b). Partial dependence plots illustrate how the predicted mean likelihood of oil palm expansion (Fig. 5a, c) and deforestation (Fig. 5b, d) change on average with distance to an informal mill and with mill extraction rate, holding all other variables constant.

Controlling for population density and the locations of villages and towns, informal mills were significantly correlated with oil palm expansion. The odds of a given land area being converted to oil palm outside industrial concessions increased by 10% (Odds Ratio = 0.91, 95% CI [0.90, 0.92]) for every kilometer decrease in distance to an informal mill (Supplementary Table 1, Fig. 4). Holding all other variables constant, land areas outside concessions within 10 km of an informal mill had at least a 50% likelihood of conversion, on average (Fig. 5a). The extraction rate of a mill was also significant, with the odds of expansion outside concessions increasing roughly 42% with every 10% decrease in the extraction rate of the nearest mill.

In contrast, the odds of expansion were 131% greater within large company concessions for every 10% increase in the nearest mill's extraction rate. On average, the likelihood of expansion inside concessions was highest near mills with an extraction rate greater than 20% (Fig. 5b). This latter finding is consistent with expectations given high extraction rates at high-efficiency agro-industrial mills (i.e., 20–22% in Southwest Cameroon). In addition to oil palm, these agro-industrial companies produce other commodity crops across their concessions, including rubber and bananas. They likely choose

to expand oil palm production in estates closest to their high-efficiency mills.

Results from the deforestation model were more nuanced. On average, we estimated a 48% increase in the odds of forest conversion outside concessions for every 10% decrease in the extraction rate of the nearest mill, indicating that forest conversion for non-industrial oil palm production was more likely near the lowest yielding mills (i.e., informal mills with the lowest extraction rates) (Supplementary Table 1, Fig. 4b). Holding all other variables constant, the predicted mean likelihood of forest conversion to oil palm increased as the extraction rate of the nearest mill decreased, while inside concessions the opposite trend was observed, with the odds of forest conversion increasing with mill extraction rate (Fig. 5d).

Additionally, the predicted mean likelihood of deforestation outside concessions never fell below 50% near informal mills, suggesting a high likelihood of non-industrial forest conversion to oil palm on average (Fig. 5b). Yet, for every kilometer decrease in distance to an informal mill, the odds of deforestation outside concessions decreased 10% (Supplementary Table 1, Fig. 4b). Taken together, these results suggest that forest conversion to oil palm outside concessions is likely, with higher odds of conversion near mills with lowest extraction rates. However, on average, the likelihood of conversion outside concessions is higher at distances of about 20 km compared to 5–15 km from informal mills.

In contrast, the odds of deforestation due to oil palm expansion inside company concessions (i.e., concessions actively clearing new land for oil palm) was 4% greater for every km decrease in distance to an informal mill (Fig. 4b). Given that agro-industries do not process FFBs from their concessions at informal mills, and we would not expect their presence to influence a company's decision to expand, this suggests that non-industrial producers may be drawn to locations where companies are actively

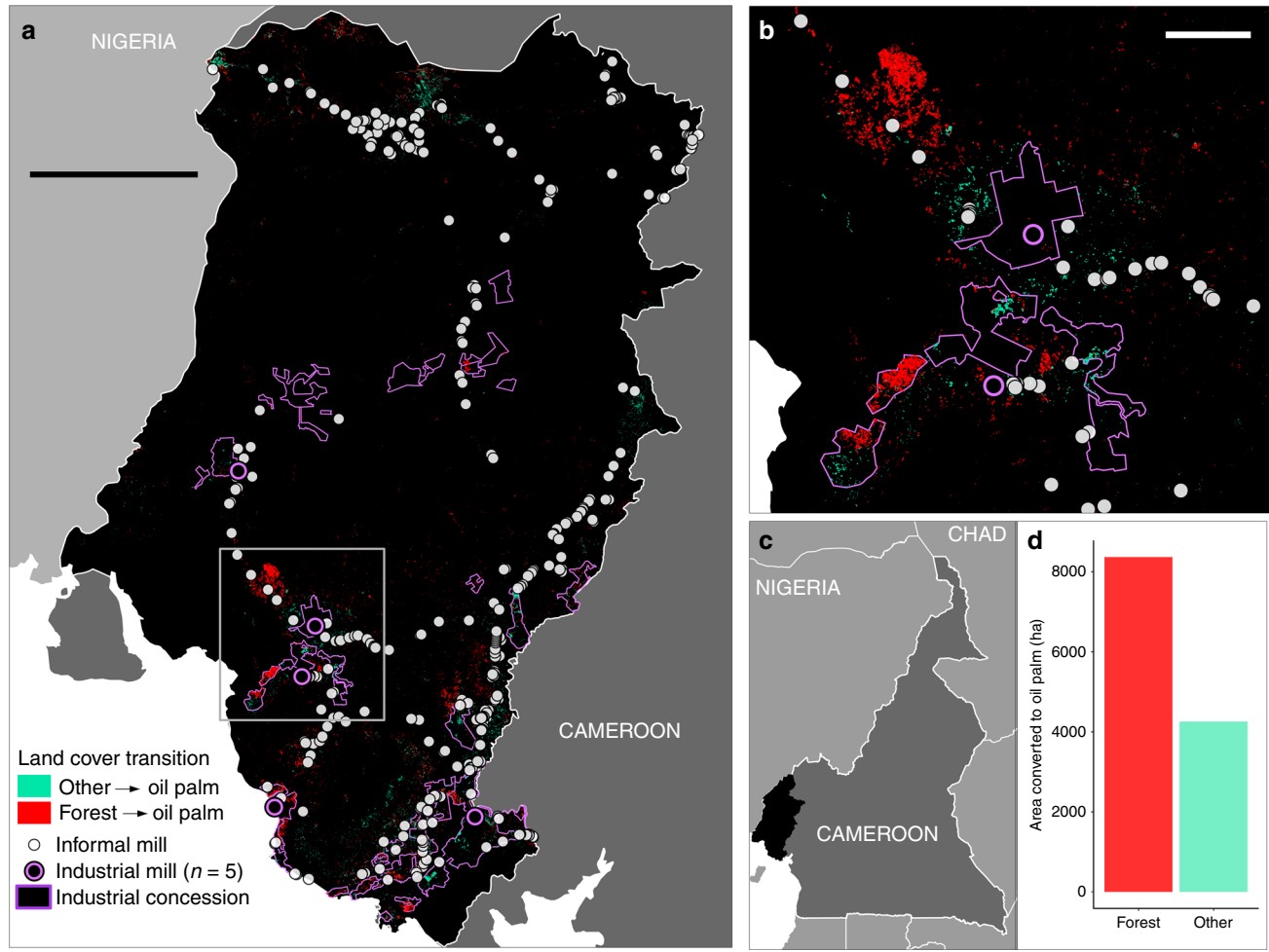

**Fig. 3** Oil palm expansion results. **a** Oil palm expansion across Southwest Cameroon from 2000–2015, with areas of forest conversion to oil palm highlighted in red. Conversion of other land cover types to oil palm are shown in turquoise. Scale bar: 40 km. **b** Most oil palm expansion occurred outside industrial concessions, outlined in purple. Scale bar: 6 km. **c** The Southwest Region of Cameroon. **d** Total land area converted for oil palm expansion coming from forest (red) and other land cover types (turquoise). The area converted for oil palm expansion was double the total area of conversion across other land cover types

expanding, and informal mills are being established there. We expand upon this idea below.

## Discussion

Rapid expansion of oil palm plantations resulting in tropical deforestation in Southeast Asia has focused global attention on the sustainability of the sector. We set out to quantify the extent of deforestation associated with non-industrial oil palm production and explore spatial relationships with high-efficiency, industrial mills vs. other milling systems in Southwest Cameroon, a top producing region of Africa. Contrary to the widely publicized narrative of industrial-scale expansion, most oil palm expansion and associated deforestation did not occur inside large, agro-industrial concessions. Instead, over two-thirds of expansion and deforestation was carried out by non-industrial producers, near small-scale, low-efficiency, informal mills. Findings from this study also demonstrate that oil palm is expanding at the expense of forests throughout the region, unconstrained by the location of company-owned industrial mills. The increased odds of expansion near informal mills and deforestation near the lowest yielding informal mills highlight the key role of a booming informal economic sector in driving rapid land use change.

Despite concerns that industrial-scale oil palm expansion threatens African tropical forests[16], we provide evidence that

non-industrial producers accounted for twice the area of expansion carried out within industrial concessions in Southwest Cameroon, contributing substantially to deforestation. While we anticipated non-industrial oil palm expansion activity, the magnitude of its role was unexpected given the competitive advantage of large companies associated with economies of scale. The observed non-industrial producer proliferation is, however, consistent with their ability to navigate a complex land tenure system[35]. Like many West and Central African countries, customary and statutory land tenure systems in Cameroon create a complex legal pluralism under which land users operate[42]. For example, foreign investment in land for oil palm production in 2009 resulted in the allocation of 73,000 ha of land by the Government of Cameroon under a 99-year lease to a private New York-based firm, Sithe Global Sustainable Oils Cameroon, Ltd. This land was already inhabited by more than 14,000 people, resulting in conflict and opposition that led to a substantial reduction in the lease size and duration[43]. In contrast, non-industrial producers from Cameroon, who are familiar with customary land tenure systems in their region, are more capable of obtaining the necessary permissions to acquire land for cultivation. The concentration of this non-industrial expansion near informal milling facilities rather than near high-efficiency, industrial mills ran counter to expected patterns based on market efficiencies associated with

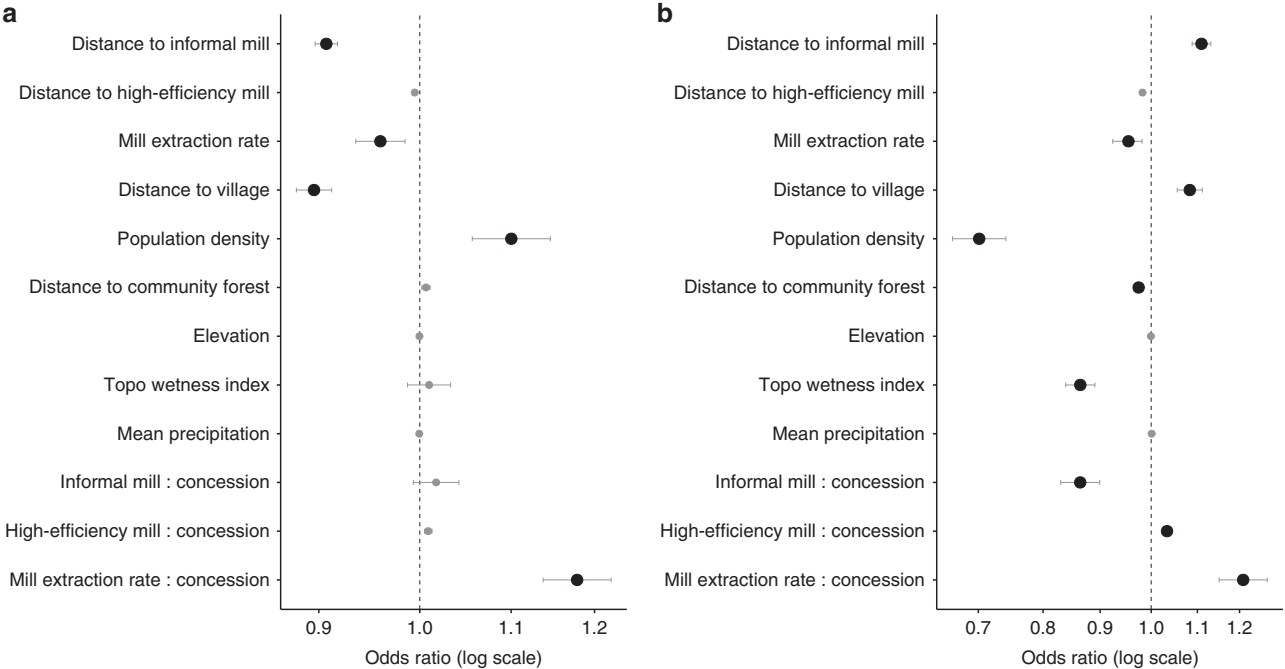

**Fig. 4** Relative influence of mills on expansion and deforestation. **a**, **b** Odds ratios from the binomial logit models predicting oil palm expansion (**a**) and deforestation due to oil palm expansion (**b**). Variables significantly affecting oil palm expansion (**a**) and deforestation (**b**) are indicated by the larger, black circles (significant at Wald test, $p < 0.05$ with odds ratios that do not overlap 1). Variables with odds ratios to the right of the dotted line at 1 indicate that an increase in that variable is associated with greater odds of oil palm expansion or deforestation, while odds ratios to the left of the dotted line indicate that an increase in that variable is associated with decreased odds. Bars correspond to the 95% confidence interval for odds ratios calculated using White–Huber standard errors from the oil palm expansion model ($n = 1,929,816$) and deforestation model ($n = 12,266$)

economies of scale. However, it is consistent with research demonstrating the important role of informal economies in sub-Saharan Africa, which account for over 60% of the continent's total GDP[44].

Non-industrial producers actively expanding and clearing forest operate across a range of production scales, supplying to a variety of milling systems that provide palm oil to local and regional consumers through complex, dispersed supply chains[35]. Although informal mills are highly labor intensive and low yielding, they appear to offer producers several benefits over the agro-industrial milling alternative, evidenced by the region's boom in informal mills since 2000. Given relatively low installation costs, particularly for manual mills, the informal milling sector in Cameroon is far more dynamic than industrial milling operations which require significant investment. Several informal mill managers reported starting with manual presses and upgrading to semi- and fully-mechanized systems. Because of this dynamism, the expansion of oil palm cultivation and mills co-occur, whereby non-industrial producers become established in areas prior to informal mills in some cases, and producers establish near existing informal mills in other cases.

In addition to our finding that non-industrial producers aggregate near informal mill operations, our results demonstrate that informal mill operations aggregate near company concessions where companies are actively clearing land explicitly for oil palm production. Agro-industrial companies exclusively use their high-efficiency mills. One possible explanation for this pattern relates to FFB theft. The companies that own actively expanding concessions have reported rampant FFB theft from their plantations, in some cases arguing that stolen FFB are supplied to nearby informal mills[39,45].

Another possible explanation relates to agglomeration benefits derived by informal mills. Informal mill operators benefit from agglomeration through improved road conditions, a higher concentration of input suppliers, and greater crop-specific agronomic knowledge and informational exchange near large company concessions. Agglomeration theory supports the clustering of mills around shared input suppliers (i.e., producers supplying FFB) if transportation costs are high and the input demand of an individual mill is not large enough to exploit the scale economies of oil palm production[28]. Poor road conditions in Southwest Cameroon and the lack of access of most farmers to transportation options meets the high transportation cost condition. The condition that input demand of an individual mill is low relative to production is supported by our observation that most informal mills operate at very low capacities. As a result, many informal mills report an inability to process the FFB production from their supply shed within the preferred 24–48 h window, particularly during the peak season. Industrial mill managers also cite a frequent inability to keep up with processing demand during the peak season[30], suggesting excess FFB supply at least seasonally.

The establishment of non-industrial producers near agro-industrial concessions may also relate to direct contractual relationships with companies, in which case a producer may acquire access to land, transportation to the company's industrial mill, high-yielding seedlings, or labor for harvesting. Because large companies face lower mill installation and processing costs relative to independent, informal operators stemming from milling-related scale economies, they should out-compete smaller mill operators. However, seasonal limitations to the benefits a producer gains from association with industrial mills appear to support a competitive advantage in small-scale milling alternatives.

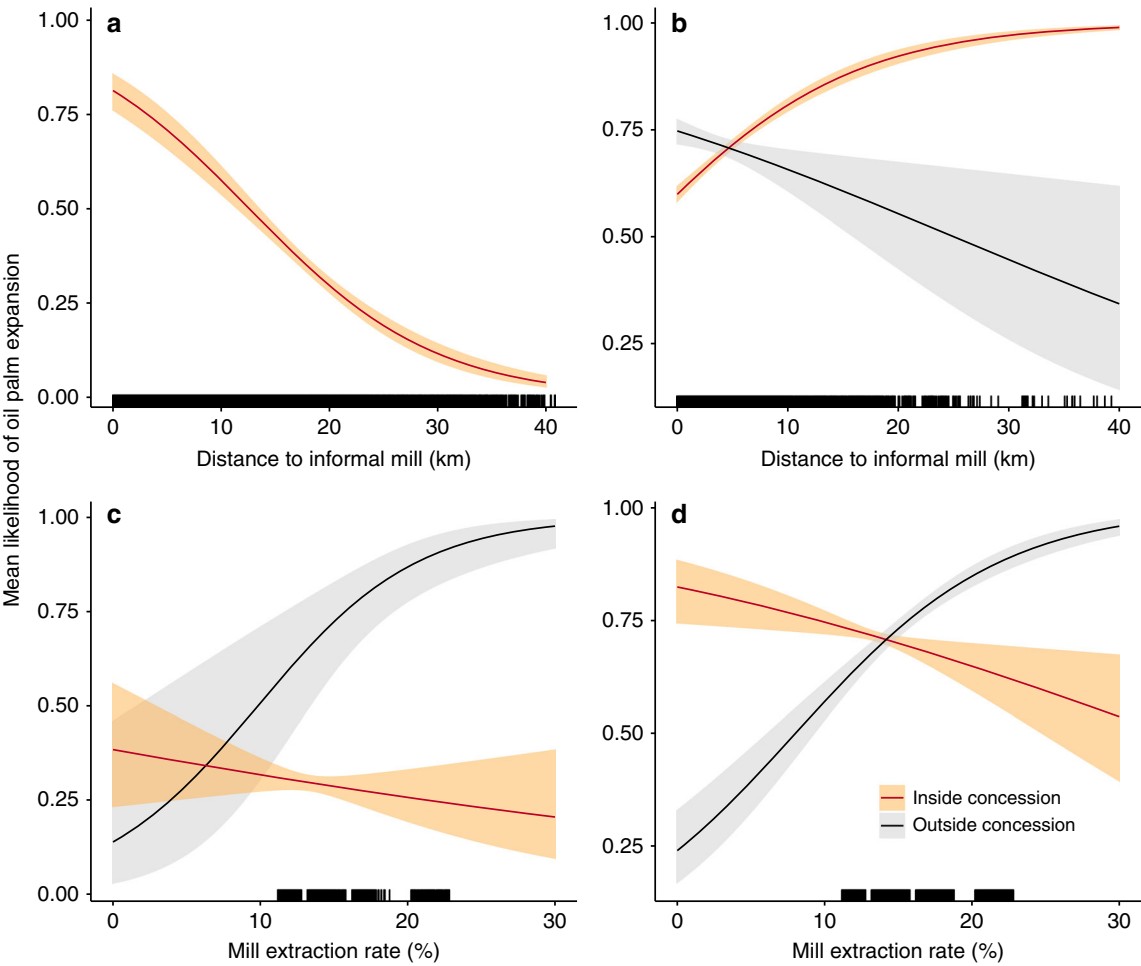

**Fig. 5** Partial dependence plots. **a**–**d** Differences in the predicted mean likelihood of oil palm expansion (**a**, **c**) and expansion resulting in deforestation (**b**, **d**) inside and outside agro-industrial concessions as a function of distance to informal mill and mill extraction rate. The predicted mean likelihood of oil palm expansion inside concessions as a function of distance to informal mill is omitted from panel **a** due to the variable's lack of statistical significance. Shaded areas around each line correspond to predicted mean likelihood 95% confidence intervals calculated using White–Huber standard errors from the oil palm expansion model ($n = 1,929,816$) and deforestation model ($n = 12,266$)

The seasonality of FFB production, palm oil prices, and transport costs have been shown to incentivize producers supplying to agro-industrial mills in Cameroon to seek alternatives[46]. Strong seasonal precipitation trends result in fluctuating palm oil market prices based on peak and low production seasons. During the low period of oil palm production (June-September), which coincides with the short dry season, regional palm oil prices can increase substantially given restricted FFB supply. Despite this, agro-industrial mills buy at a fixed price, making them less competitive than informal mills whose oil prices are allowed to fluctuate between the low and peak seasons based on supply[39,46]. Producers and traders often retain palm oil produced during the peak season to sell during the low season, when prices are higher. Supplemental income from this type of storage is not possible when a producer sells their FFB to a high-efficiency industrial mill, since they do not recuperate any palm oil to be sold as a value-added product[35]. Additionally, as oil palm cultivation has become increasingly widespread, many producers face difficulties transporting their FFB to the five industrial mills concentrated in the southern third of the region, particularly during the rainy season on unpaved roads.

It is likely that some producers supply to both industrial and informal mills, depending on seasonal processing demand,

transportation costs, and palm oil prices[46]. This needs to be further researched. Whether producers are seasonally taking advantage of the informal milling sector or abandoning their relationships with industrial mills altogether, high local and regional demand for red palm oil ensures a relatively stable and persistent market for palm oil produced by informal mills. Non-industrial mills also provide a major source of income to rural households, with job opportunities for both skilled and unskilled workers, including positions held by women, a feature absent from most industrial-scale operations[31,37,47].

While these mills offer additional benefits for non-industrial producers, including an alternative source of income and greater autonomy, our findings shed light on the need to account for this informal economy when developing sustainable palm oil sector pathways. We found widespread expansion among non-industrial producers. This expansion, in combination with an increased need for milling alternatives, has led to a rapid rise in informal mills. We demonstrate that the proliferation of informal mills is buttressing oil palm expansion and associated deforestation throughout the region by reducing a producer's geographic tie to the handful of existing high-efficiency industrial mills.

Unfortunately, the increase in cultivated area by non-industrial producers is unlikely to satiate increasing regional demand for

palm oil and will likely result in further expansion given low on-farm yields combined with the limited capacity and low extraction rates of informal mills. Furthermore, our finding that non-industrial producers are actively expanding and clearing forest near low-efficiency, informal mills highlights the need to consider this informal sector in future sustainability solutions. By contrast, palm oil processing in Southeast Asia is dominated by industrial milling due to the relatively recent introduction and rapid modernization of the oil palm industry that began in Malaysia in the 1960s and expanded to Indonesia[1]. Because both large and small-scale producers in Malaysia and Indonesia rely on industrial mills for processing, corporations attempting to eliminate deforestation from their supply chains have identified these mills as a key leverage point for regulating the land use practices of their suppliers. Although, comprehensively identifying all the producers who supply to each mill and tracking associated deforestation remains a challenge. In Southwest Cameroon, we found that industrial mills represent a mere 1% of all processing facilities in the region and only 28% of the regional processing capacity. Given the widespread use of informal mills, tracking deforestation through mill certification becomes an even greater hurdle, suggesting that current, non-state palm oil sustainability approaches (e.g., RSPO) targeting industrial mills and their supply shed as a certifiable unit will fall short in this context. Additionally, many of these unregulated, low capacity mills have poor environmental management (e.g., effluence) and health standards (e.g., high free fatty acid content).

Our results point to the widespread use of non-industrial milling systems for oil palm production that is common across producing regions in Africa (e.g., Ghana, Nigeria, Libera), owing to the long history of production and consumption[1,24]. Given concerns about future oil palm expansion moving to Africa[1,16], improved understanding of existing expansion dynamics offer insights into regionally relevant strategies for avoiding large-scale deforestation associated with commodity crop expansion. Although this study focused on Cameroon's top producing Southwest region, oil palm has been produced and consumed widely across West and Central Africa for thousands of years[24]. Further research is necessary to verify whether our results are generalizable to other regions of Cameroon, and other African countries. However, our findings are generally consistent with recent research indicating that small-scale agricultural expansion has become the dominant driver of deforestation and disturbance in Congo Basin countries, doubling between 2000 and 2014[48].

Far from a static system, the dynamic and evolving oil palm sector in sub-Saharan Africa presents unique sustainability challenges and opportunities. Findings from this research shed light on important land use change differences in the sector relative to its Southeast Asian counterparts, stemming from small-scale expansion associated with a dynamic informal milling system, linked to local and regional supply chains. Although our results highlight a connection between informal mills and unsustainable forest clearing, widespread use of these mills with low capacities and extraction rates presents a considerable opportunity to prevent further deforestation and increase palm oil yields at the processing stage and achieve major production gains from the existing cultivated area. High per capita consumption and rising palm oil demands in sub-Saharan Africa spotlight the need to consider these differences when identifying regionally relevant sustainability pathways.

## Methods

**Southwest Cameroon study region**. We mapped oil palm expansion in Cameroon's top-producing Southwest Region. We restricted the analysis due to persistent, heavy cloud cover during the study period in the Littoral and Central Regions, preventing a comprehensive remote sensing analysis. In the Southwest, we limited our analysis to the humid forest zone agro-ecologically suitable for oil palm production according to Cameroon's Research Institute for Agricultural Development[32]. Across the study region, elevation increases from sea level (0 m) to more than 4000 m above sea level at the summit of Mount Cameroon. Mean annual precipitation averages 2900 mm, ranging from ~2100–3500 mm[49]. Rainfall is distributed during a short, wet season from September until late November and a long, wet season from late March until June, with air temperatures averaging ~23 °C annually. Estimates of average aboveground biomass in the region range from 209–300 Mg C ha$^{-1}$[11,50,51].

**Pixel composites and image processing**. Oil palm expansion across Southwest Cameroon was mapped from 2000–2015, a period of national growth in production and area expansion of the crop (Fig. 1). To classify oil palm, forest, and other land cover, we used 30 m resolution Landsat satellite data. Remote sensing analyses in Cameroon are severely limited by the amount of available imagery due to persistent cloud cover. To overcome this issue, we created pixel composites using a cloud-scoring algorithm developed in Google Earth Engine to identify the most cloud- and cloud shadow-free pixels within the selected study region and period. We first filtered Landsat 5, 7, and 8 orthorectified sensor-radiance and top-of-atmosphere reflectance (TOA) image collections by dry season date and year, obtaining scenes between 1 December –28 February for the years 2000–01 and 2014–15, which we refer to as 2000 and 2015. Landsat 8 imagery was also filtered using 16-bit Quality Assessment (QA) band bit-packed information on quality conditions associated with clouds, cirrus clouds, water, snow, ice, and terrain occlusion. To avoid misinterpretation from bit unpacking and bit shifting, we masked the following QA band values that are known to occur regularly and based on occurrence and visual association with cloud contamination in the study region imagery: 36864, 49152, 49184, 52224, 52256, 53248, 56320, 57344, 59424, and 61440.

Each pixel was scored across all scenes within the filtered date-range based on the presence of cloud and cloud shadow contamination, defined by physical properties. Due to the bright and cold nature of clouds, all types of clouds should have Band 7 (SWIR-2, ~ 2.1–2.3 μm) TOA reflectance larger than 0.03 (drawn from LEDAPS internal cloud masking algorithm) and thermal infrared (10.40–12.50 μm) values <27 °C (drawn from ACCA)[52]. All bands were used in cloud and cloud shadow scoring. Scores were linearly rescaled to normalize values between 0–1. The algorithm selected sensor-radiance data identified as having the lowest TOA cloud and cloud shadow score on a pixel-by-pixel basis, resulting in nearly cloud free composites for each year.

Sensor-radiance was calibrated to apparent surface reflectance using the Carnegie Landsat Analysis System - Lite (CLASlite) software[53]. Automated Monte Carlo Spectral Unmixing was also carried out using CLASlite to calculate fraction cover information on photosynthetic vegetation, non-photosynthetic vegetation, and bare substrate to provide additional information on small biophysical variations at the sub-pixel scale. These fractional covers are determinants of vegetation composition, structure, biomass, physiology, and biogeochemical processes. As such, they enable detection of differences between agricultural systems and forest with greater sensitivity. Prior to classification, a water mask was created and applied to all data using ENVI.

**Oil palm classification**. Random Forest models (hereafter RF) were used to classify imagery in 2000 and 2015. Input data in the RF models included surface reflectance, sub-pixel information on fractional vegetation cover, topographical and texture data, and three vegetation indices (NDVI and Tasseled Cap greenness and wetness). The RF models were trained to classify the following land cover types: forest, mature oil palm, immature monoculture, and other land cover (Fig. 2). Forest was defined as tree cover >50% over an area of at least 0.09 ha (30 m) and included primary and secondary forests. Other land cover included mixed-cropping systems, mature monoculture systems other than oil palm (e.g., banana, rubber), and other non-forested, non-oil palm land cover types (e.g., bare soil and urban areas). Classification was conducted using the randomForest package in R.

To resolve oil palm planted area in the imagery, classification training data consisted of geo-referenced oil palm fields identified from field visits to agro-industrial concessions and non-industrial plantations throughout the region in 2015. This information was supplemented with land cover identification using DigitalGlobe data from the Mapbox Editor for the year 2015 and high-resolution imagery from Google Earth for years circa 2000 and 2015. In this study, mature oil palm refers to plantations that are ~4 years or older, after which point palm fronds have developed and the canopy is closed[54,55]. The spectral separability of mature oil palm plantations from other land cover types in the region enabled classification with reasonably high accuracy (see Supplementary Table 2).

Immature oil palm plantations, within the first 3 years after planting, largely consist of palm trees with fronds still under development, and the canopy is not yet closed. Given the high fraction of bare soil and the lack of spectral separability between immature oil palm and other monoculture cropping systems in this early stage of growth, we were unable to detect immature oil palm monoculture systems as a unique class with sufficient accuracy. Thus, immature monoculture systems for different monoculture crops (oil palm, rubber, and banana) were grouped into a single class. Image classification was conducted using the randomForest package in R. One-third of the training data were randomly selected and withheld from the

model fitting process and used as a test dataset to estimate classification accuracy. Area weighted accuracies were calculated using methods described in Olofsson et al.[56]. Model and class-level accuracy information for each classification is described in Supplementary Methods and Supplementary Table 2.

The immature monoculture class in the Southwest Region of Cameroon is comprised of one of three crop types: oil palm, banana, or rubber. Owing to the high fraction of immature monoculture pixels in 2015, and because immature mono-cropped oil palm was not spectrally distinguishable from immature mono-cropped banana or rubber, we relied on a manual, visual interpretation method to estimate the fraction of pixels classified as immature monoculture that were immature oil palm. We estimated this fraction by sampling ten percent of all pixels characterized as forest or other in 2000, and immature monoculture in 2015 ($n = 23,347$). We visually identified whether these pixels classified as immature monoculture in 2015 were immature oil palm using high-resolution imagery from Google Earth and DigitalGlobe data from the Mapbox Editor. Despite being spectrally indistinguishable from other monoculture systems using Landsat data at 30 m resolution, we found that detection of immature oil palm plantations using the human-eye was straightforward at high spatial resolutions (roughly ≤5 m).

Using this approach, we estimated that 64.56%, 95% CI (64.55%, 64.57%) of the randomly sampled pixels that transitioned to immature monoculture in 2015 were immature oil palm. In contrast, 35% were either another type of immature monoculture system (banana or rubber) or were misclassified. Of the area forested in 2000 that transitioned to immature monoculture systems, 92.78% was a transition to immature oil palm, while 7.22% was a transition from forest to some other type of immature monoculture system. Based on this estimated fraction of immature oil palm within the immature monoculture class, 45.26% of land area converted to oil palm in 2015 was mature oil palm and 54.74% was immature oil palm.

**Change detection.** Rather than relying on the two classified maps to detect areas of oil palm expansion from 2000–2015, which can lead to error propagation, we used a hybrid approach that combined image differencing with post-classification results to characterize from-to land cover transitions[57]. Image differencing provides a method for detecting biophysically relevant vegetated areas that underwent change. To conduct image differencing, we quantified changes in five spectral indices from 2000–2015: Tasseled Cap greenness and wetness, and the three sub-pixel bands on fractional vegetation cover. A change detection threshold at ±2 standard deviations around the mode was defined based on expert knowledge and previous studies[58]. Pixels were included if they exceeded this threshold for at least two indices.

This image differencing analysis identified locations of biophysical change across the landscape, resulting in a dataset of change pixels fewer in number compared to pixels identified as undergoing a transition based on a comparison of the classification maps alone. Within these change pixels, classification maps for 2000 and 2015 were used to determine the land cover transition from 2000–2015. Forest conversion to oil palm (i.e., deforestation) was defined as a stand-replacement disturbance, with detectable forest degradation[59,60] included in this definition. Any change detection pixel that underwent a transition from forest to mature oil palm monoculture was considered deforestation due to oil palm expansion.

**Mapping palm oil mills.** To create a spatially explicit dataset of palm oil processing mills in Southwest Cameroon, we mapped all mills in the region using GPS points during multiple field visits in 2015–2017, totaling 498 mills. For each mill, we collected data on type (manual press, semi- or fully-mechanized wet digester, and semi- or fully-mechanized dry digester), capacity (tons FFB hour$^{-1}$), extraction rate (%), and the year of establishment (Supplementary Figs. 1–2).

**Socioeconomic and agroecological control variables.** We compiled data for 11 covariates based on factors we expect to be associated with the likelihood of oil palm expansion and the likelihood of forest conversion for oil palm expansion. These include market access, indicators of agroecological suitability, and current land allocation (Supplementary Table 3).

Oil palm producers' point of entry in the palm oil market is via mills. We included three mill covariates, two of which were measures of distance (km) distinguishing between high-efficiency industrial mills, and low-efficiency informal mills. Given the heterogeneity among informal mills, we also included extraction rate (%) as a proxy variable for mill type. Oil extraction rate is a widely used metric of mill performance within the oil palm industry relating to milling productivity. It is the percentage of the weight of palm oil recovered from a known weight of FFB processed. Thiessen polygons were used to construct a defined area of influence around each mill. Each Thiessen polygon was assigned the value of the mill extraction rate. The highest extraction rates correspond to mechanized, agro-industrial mills (20–22%), while manual mills have the lowest extraction rate (12%). In sub-Saharan Africa, an oil palm producer's market access can also be influenced by their connection to informal supply chains, whereby producers sell directly to local consumers[35]. These local supply chains are concentrated in urban and rural settled areas. Thus, we also included distance to the nearest town or village (km) and population density (people ha$^{-1}$, log transformed) as measures of market access, drawn from WRI[61] and AfriPop[62] respectively.

Locations within agro-industrial concessions were identified using data from WRI[61]. These data do not specify crop type and thus broadly refer to concession areas leased by agro-industries. Within Southwest Cameroon, they are used for cultivating rubber, banana, or oil palm. Distance to a community forest was included to control for the influence of land allocated by the government of Cameroon to communities within the non-permanent forest domain[61]. Beyond this high-level land allocation information, a lack of data made it impossible to include customary land tenure in the models, although we recognize that complex land tenure systems in the region likely influence land use decision-making[63,64].

Finally, mean annual precipitation (mm yr$^{-1}$), elevation (m) and topographic wetness index (unitless) were included as measures of agroecological suitability. Rainfall patterns, an important determinant of FFB yields, vary across the Southwest Region, with some coastal areas averaging over 3000 mm yr$^{-1}$. Producers in Cameroon report locations with high rainfall, but less persistent cloud cover as ideal for achieving higher on-farm FFB yields. Across the tropics, lower elevation areas tend to be most susceptible to agricultural expansion given their greater accessibility[65]. We also included topographic wetness index (TWI) as an agricultural suitability measure, corresponding to a location's hydrological propensity for inundation (higher values) or runoff (lower values). Elevation and TWI were estimated using Shuttle Radar Topography Mission (SRTM) data[66], and mean annual precipitation was calculated using CHIRPS[49].

**Oil palm expansion and deforestation models.** Using results from the change detection analysis, we conducted two spatial analyses to evaluate the effect of the above variables on oil palm expansion. First, we developed a binomial logit model to estimate oil palm expansion across the entire Southwest Region. In areas where expansion took place, we then modeled deforestation due to oil palm expansion relative to expansion at the expense of other land cover types, for example replacing a rubber plantation or mixed crop farm. We fit two separate logit models. The first regression estimated oil palm expansion using all expansion and non-expansion grid cells ($n = 1,929,816$). The second estimated deforestation due to expansion, relative to the conversion of other land cover types, on a reduced dataset that included only grid cells where expansion occurred ($n = 12,266$). Due to the rarity of expansion at the landscape scale, the expansion regression was fit using a weighted loss function. A weight variable was calculated by summing all no-expansion pixels and all expansion pixels and assigning no-expansion pixels a weight of the sum of expansion pixels divided by the sum of all pixels included in the analysis. Expansion pixels were assigned a weight equivalent to the sum of no-expansion pixels divided by the sum of all pixels in the analysis. The weight variable was included in the model by assigning the calculated weights within the glm function in R.

Both models were conducted at a 1 ha grid cell unit of analysis ($100 \times 100$ m), which was chosen to provide results at a regionally relevant scale, where farms range from one to thousands of hectares. To distinguish between milling effects on expansion occurring inside agro-industrial concessions compared to outside, we included a concession dummy variable with locations inside concessions equal to one. By interacting the concession dummy with distance to high-efficiency industrial and low-efficiency informal mills and mill extraction rate, we estimated their effects separately on land use change inside vs. outside an agro-industrial concession. This allowed us to identify differences in milling effects on oil palm expansion and deforestation depending on whether the expansion was carried out by agro-industries or non-industrial producers. Correlation analyses were performed between explanatory variables to ensure no collinearity existed in the dataset (Pearson's $r < 0.5$, Supplementary Fig. 3).

Both the oil palm expansion and deforestation non-spatial binomial logit models (Supplementary Table 4) yielded error terms with high spatial autocorrelation. Moran's $I$ was estimated at 0.439, relative to an expected value of $-0.001$ ($\sigma^2 < 0.0001$, two-tailed, Student's $t$ test, $p < 0.0001$) for the non-spatial oil palm expansion model, and 0.102, relative to an expected value of $-0.003$ ($\sigma^2 < 0.0001$, two-tailed, Student's $t$ test, $p < 0.0001$) for the non-spatial deforestation model. To address the issue of spatial autocorrelation, autocovariate regressions were fit by including a spatial autocovariate term in each model. Dormann[67] and Dormann et al.[68] put forth the claim that autocovariate models can underestimate the effects of important parameters and generate bias in the estimates. Bardos and coauthors demonstrate, however, that this is not the case when a valid neighborhood weighting scheme is employed[69]. We used the spdep and lattice packages in R to calculate neighborhood weights using the method described in Bardos et al.[69]. Parameter estimates from the spatial autocovariate regression were comparable to the non-spatial models (Supplementary Table 4).

Model parameters and accuracy were estimated using 10-fold cross validation for the non-spatial and spatial autocovariate model. Robust (White–Huber) standard errors were calculated using White's estimator via the foreign and sandwich packages in R. Results are reported from the spatial autocovariate oil palm expansion ($n = 1,929,816$) and deforestation ($n = 12,266$) models. To evaluate performance of the overall model, prediction error was calculated as the average area under the receiver operating characteristic (ROC) curve (AUC) across all cross validation test sets using the ROCR package in R (Supplementary Fig. 4). Because the purpose of this analysis was inference, as opposed to prediction, the AUC values reported in the Supplementary Methods and Supplementary Fig. 4 demonstrate sufficiently reasonable performance of both models. We focus on

inference related to the significance of specific milling variables of interest in the model, and their relationships with oil palm expansion and deforestation.

## Data availability

Source data for the remote sensing analysis came from the SRTM 30 meter elevation data[66] and data from the NASA Landsat satellite archive, available from the USGS Global Visualization Viewer (GloVis) website (https://glovis.usgs.gov/), or the Google Earth Engine API (https://code.earthengine.google.com/). The land cover classifications and the land cover change map were deposited in the Stanford EarthWorks public data repository under the following accession codes: 'https://doi.org/10.18735/4nse-8871', 'https://doi.org/10.18735/952x-w447', 'https://doi.org/10.18735/0mw1-qq72'. Mill data used in the spatial models are available aggregated at the Division (municipality) scale from the corresponding author on reasonable request. All additional data sources used in the spatial modeling analyses, listed in Supplementary Table 3, came from the WRI Forest Atlas of Cameroon (http://cmr.forest-atlas.org/), WorldPop (http://www.worldpop.org.uk/), and the CHIRPS rainfall dataset (http://chg.geog.ucsb.edu/data/chirps/).

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

## Acknowledgements

This work was supported by the National Science Foundation Graduate Research Fellowship Program (Grant No. 2012118590). The Stanford Global Development & Poverty Initiative, Morrison Institute for Population and Resource Studies, Stanford Center for African Studies Graduate Fellowship Program, and a McGee-Levorsen Research Grant provided additional funding for fieldwork. Field support was provided by the Center for International Forestry Research.

## Author contributions

E.M.O. conceived and coordinated the study; R.N.N led the mill mapping; E.M.O. conducted all analyses; E.F.L. and R.L.N. supervised and supported the research; E.M.O. carried out fieldwork to ground truth the remote sensing analysis and wrote the manuscript; All authors participated in discussion and contributed to the editing of the manuscript.

## Additional information

**Competing interests:** The authors declare no competing interests.

