## [Peer Review File · Nature Communications]

Reviewers' comments:

Reviewer #1 (Remarks to the Author):

The paper entitled 'Oil palm expansion at the expense of forests in Southwest Cameroon associated with proliferation of informal mills' examines the role of informal artisanal mills in oil palm expansion and associated deforestation in Southwest Cameroon. The main claims are:

1. Oil palm expansion in Southwest Cameroon occurs in relation with small-scale informal mills.
2. 2/3 of expansion and deforestation associated to oil palm plantations between 2000 and 2015 was carried out by non-industrial growers.
3. Deforestation is happening, even in this densely populated area of Cameroon.

Studies on oil palm artisanal mills have so far been focusing on technical and economic aspects, but not on land-use change dynamics. Reference is made to most of previous publications on this subject in Cameroon. This study thus addresses artisanal milling from a new and interesting angle.

The method is well described and adapted to the objective of the research. The statistical analysis is appropriate. Results are clearly exposed. The discussion could benefit from more references and comparison to other studies. In the introduction, previous literature on industrial agriculture in Central Africa and its impact on forests could be deepened.

Land use change analysis focusing on oil palm expansion and associated deforestation at the national and sub-regional scale would be an interesting follow-up to this study. This could be crossed to industrial concession locations to get an idea of the genericity of the results. Of course a repetition of the whole method including field work to characterize artisanal milling would be great, but would require major means and time.

I propose below some comments and suggestions on each part of the paper in the hope that they might help strengthening the research.

Introduction:

References 16 to 20 (quoted L 73-74): the reader would expect references on African cases in this paragraph, on the five references quoted here only one refers to Africa, and it is an NGO report not a scientific publication. Suggested references: Deininger et al 2011, Megevand 2013, Feintrenie 2014, Oyono et al 2014, Marquant et al. 2015, Feintrenie et al 2016.

"we explore whether trends in sub-Saharan Africa conform to the widely held view that large, industrial companies drive most oil palm expansion." (L 76) To explore this hypothesis, a study at a national or sub-regional scale would be useful. At the study site scale, the age of industrial concessions and the level of land saturation and availability to conversion to agriculture (not under a conservation status or allocated to other purpose such as mining or urban development) are decisive to current dynamics of plantation expansion.

L84: Is it where the introduction finishes and the 'Results' begin? There is a lack of clarity in the plan of the paper.

Results:

The Southwest region "is 86% forested" (L 162), how much of this is suitable to oil palm? What definition of 'forest' is used here?

A short historical background on the 'two agro-industrial companies' (L 164) would be important to understand their current dynamic of expansion. The map covers three companies, two with a long history in the region, and one quite new.

L 165 'companies' is used for mills, whereas L 164 it is used for industrial firm. The use of this word needs to be consistent.

"expansion occurred outside industrial plantations" (L 181), 'concessions' would be more appropriate than 'plantations' here, same on L 203, 205, 216, 220, 222, 233 and 237 (Figure 5 title).

Figure 3: Hansen et al (2013), used for the tree cover data in figure 3 (L 192), does not make distinction between natural forest and plantations (including rubber or oil palm plantations). Are land use transitions shown on the figure conversion from natural forest or annual crops/savannas to oil palm, or conversion from natural forests and tree plantations or annual crops/savannas? Is land use change in the CDC concession in Bamuso deforestation of natural forest or conversion from rubber plantations? In the method section (L 366-401), land cover analysis is detailed and does not include the support of Hansen et al (2013) classification. Could you clarify the use of Hansen et al. (2013) tree cover data in figure 3?

How do you consider land availability in the analysis? Isn't it an explicative variable of oil palm expansion or deforestation? Land availability is related to population density, but not only. Did you exclude from the land-use analysis land that were available to oil palm conversion because of their status (protected, public reserve for industrial or urban development, etc.) or tenure?

Discussion:

L 245-252 are repetitive of introduction and results.

History is referred to but not enough used as explanatory factor of the present location of oil palm plantations and expansion:

- "We expected to find that economies of scale would lead to greater expansion of large-scale plantations" (L 259). Quite a strange hypothesis knowing the history and age of most of the industrial concessions in the study site, with the exception of SG-SOC concessions.
- "concentration of non-industrial expansion near informal milling facilities rather than near high-efficiency mills." (L263) Why is this surprising in the Cameroonian context? Informal mills develop along the roads, nearby or within villages, and where oil palm small scale plantations are already developed at short distance. Their presence might foster the expansion of plantations if there is enough land available. Is there enough land available to expand oil palm plantations around industrial mills and plantations?

“informal mill operations aggregate near company plantations” (L 282), no explanation regarding thefts of FFB from industrial plantations? This is a common complaint of industrial plantations managers in the area...

“The establishment of non-industrial producers near plantation estates also relates to direct contractual relationships with companies” (L 301). Are new contracts signed between companies and smallholders in recent days in the study site? Can this factor explain present expansion of small-scale plantations?

L310 and following: it would be good to link rainy seasonality to seasonality of prices for the non-informed readers.

“Our results suggest that current palm oil sustainability approaches (e.g., RSPO) that target company mills and their surrounding supply shed as a certifiable unit will fall short in this context.” (L339) Which result does suggest this? No reference to support the statement?

It is sad that the discussion doesn't give any clue to the genericity of the results. To Cameroon? To Central and West Africa? To non-african producing countries? The last paragraph of Discussion quotes Southeast Asia and sub-saharan Africa, but no literature is quoted to discuss a comparison.

Laurène Feintrenie

Forests and Societies research unit

CIRAD, Univ Montpellier, France

Associate scientist to CATIE and to ICRAF

CGIAR Research Program on Forests, Trees and Agroforestry

CATIE, Turrialba, Costa Rica.

Reviewer #2 (Remarks to the Author):

Summary

Destruction of rain forests for the production of palm oil has been a strong driver of biodiversity loss. While most of the oil palm-related deforestation since 2000 has taken place in Southeast Asia, there is increasing concern about sustainability issues around new oil palm frontiers in Africa and South America. This manuscript by Ordway et al. presents new and relevant insights into oil palm

expansion dynamics and related deforestation in a part of Sub-Saharan Africa. The authors have conducted a spatially explicit analysis of oil palm expansion between 2000-2015 in the Southwest province of Cameroon. Using Landsat satellite imagery, they measured the contribution of oil palm expansion to deforestation in and around industrial concessions. Furthermore, they analysed the relationship between different milling systems and expansion using the first georeferenced dataset of palm oil processing facilities for the region.

Using this case study of a small palm oil producing region in Cameroon, they argue that non-industrial palm oil producers – smallholder farmers and small-scale palm oil millers – are key players in palm oil production and associated deforestation. Two thirds of expansion and deforestation was due to non-industrial oil palm farms. Not only do these findings shed light on key differences in the dynamics of land use change relative to the Southeast Asian experience, it also shows that this type of small-scale production deserves more attention from civil society and researchers working on deforestation issues. Another important contribution of the manuscript is the finding that new oil palm plantations in the region are predominantly replacing forests (67%), something which has not been shown before.

The emphasis on the presence of milling facilities of the manuscript is a welcome addition to the literature. Milling is a critical step in palm oil production and considering oil palm development pathways should include consideration of this factor.

A particular strength of the study is the georeferenced dataset of all the 498 palm oil mills in the region – ranging from tiny mills with a manual press, to factory-like mills belonging to industrial plantation companies – which has been collected during multiple field visits and contained data on various aspects of the mills.

This manuscript argues rightly that the widely publicized narrative of industrial-scale oil palm expansion should be revised: at least in this region, most oil palm expansion and associated deforestation occurred outside large plantations. Hence, there needs to be increased attention to the potentially harmful effects of NGO and government schemes promoting smallholder oil palm farming; ironically, these are sometimes touted as a tool to improve sustainability. In addition, the manuscript points out that major sustainability gains could be made by improving the extraction rate of the small informal mills.

Overall, I believe that the manuscript is a convincing study of a previously neglected aspect of oil palm development. It is well structured and clearly written. The work has a number of implications for our understanding of deforestation by non-industrial palm oil producers; this is relevant not only to other researchers but also to governments and civil society.

I do, however, have some concerns about the presentation and interpretation of results, see details below.

MAJOR POINTS

1. How confident are the authors that the large amount of aggregated forest-to-oil-palm conversion in the top left corner of figure 3b is not actually part of a recent expansion by Pamol (a company)? The pattern looks much more like an industrial plantation than smallholder expansion.

The WRI Forest Atlas which was used to delineate plantation areas is a citation from 2012. On the interactive atlas online it was not easy to find details on whether this map is regularly updated and if not, what year the plantation boundaries were measured. If the map was updated in, say, 2010, this study misses five years of company expansion. Expansion certainly took place in that period, although I do not have exact information on its location. The nearest company mill is theoretically near enough to the area mentioned above for it to be used for a new plantation. It would be good for the authors to clarify this and present some evidence that this area is indeed smallholder expansion and not a new industrial plantation.

2. The method for validating immature oil palm monoculture pixels is not very clear and should be rewritten for better understanding. The authors state: 'Ten percent of all pixels characterized as conversion from forest or other land cover type in 2000 to immature monoculture in 2015 were randomly selected for validation (n = 23,347).' Considering that it appears that the increase in oil palm from 2000 to 2015 was mostly attributable to immature and not mature plantation (see supplementary material), the issue of how exactly the immature plantation was classified and mapped is very important. Therefore, please include in the methods:

a. a description of the method of classification and mapping for the remaining 90% of immature monoculture pixels

b. information on what proportion of the oil palm in 2015 was immature vs mature

c. information on what proportion of the area which was forested in 2000 but immature monoculture in 2015 was immature oil palm versus immature other crops

3. I had some difficulty understanding some of the results, and wonder if the Y axis labels in Figure 5 are mis-labelled. As I understand from the methods, these plots show the odds ratio – the probability of deforestation for oil palm, relative to the probability of deforestation for other land covers. A probability of 1 for deforestation far from informal mills, would therefore mean that deforestation for oil palm is as likely as that for other land covers. If this interpretation is correct, the

axis titles should be edited, and whether or not it is correct, the meaning and calculation of odds ratios in this paper needs to be better explained. If interpreted as a probability of 1, as the axis title indicates, it would imply that all forest pixels were deforested, even tens of km from informal mills, which is not credible. Please check and relabel as appropriate.

4. The authors' dataset indicates that informal mills are mostly very small in capacity (Figure S1). Given this, and the fact that FFB can be supplied to industrial mills from tens of km away, it is still plausible that deforestation for oil palm a short distance from plantations is partly to supply industrial mills. Lack of deforestation in the immediate vicinity of these industrial mills could be explained by past land use conversion, limiting the options for new oil palm. More caution in attributing land-use change to informal mills may therefore be warranted. In particular, a comparison of the total capacity of the five industrial mills, and how that has changed through time, with the total capacity of the informal mills, would be useful.

5. While the authors have indicated that mill data will be made available on request, in this day and age it is reasonable to expect such data to be made available in a more permanent and organised way through an online repository. If there are ethical concerns about making the mill data publicly available, this may not be appropriate, but if not, it would be good to see the data provided through a repository.

MINOR POINTS

Line 66 – 72: This information on current sustainability efforts feels tangential rather than essential.

Line 90 – 94: The manuscript sets up its hypothesis in the context of the theory of 'economies of scale'. While the findings are presented as surprising in relation to this hypothesis, the discussion mentions some very clear reasons why in the Cameroonian case study the economy of scale argument was unlikely to be valid. Therefore, it may not be necessary to include this economies of scales argument and it could be removed for the sake of brevity.

Line 151: Figure 1, it would be interesting to be able to see how this local yield relates to yields and yield improvements over time across the planet.

Line 162: The study encompasses 40% of national palm oil production. While the region chosen is a key area for both industrial and smallholder oil palm, the Littoral region is of great importance too. Cameroon's only privately owned large industrial company group is located in the Littoral (the two plantation companies mentioned in this study are part state-owned). This company is run rather differently than the two in the Southwest and so one could hypothesise that the relationships with smallholder farmers would be very different. If true, this could affect the expansion dynamics around the larger company milling facilities and thus the conclusions of this study. Perhaps something along these lines should be noted in the discussion. At minimum, the methods should state why the Southwest was chosen over other regions.

Line 190: Figure 3:

In the legend, where it states plantation, presumably industrial palm oil plantation is meant. This should be clarified. Are these only oil palm plantations or is e.g. rubber, banana etc also included? Why are smaller areas planted with oil palm not also called a plantation? What is the size limit above which it is called plantation, or what other definition do the authors use? In Cameroon, oil palm smallholdings entities can be up to 500 hectares large (often owned by rich city dwellers, run like a mini industrial plantation).

Line 281 – 286: A factor which should be included here in relation to the potential reasons for informal mills agglomerating around newly developed parts of industrial plantations is the widespread practice of stealing fresh fruit bunches (FFBs) from company land (see for example Ajambang and Ijang 2016). The presence of informal mills close to newly developed oil palm within company plantations rather than older areas within the plantations may well have to do with the fact that it is easier to steal FFBs from younger palms because they are shorter and are typically located further from the main company infrastructure.

Lines 316-318: Description of the issues surrounding seasonal changes in price and production is not very clear. Rather than write "Due to supply chain configurations, supplemental income from this type of storage is not possible when a producer sells their FFB to a high-efficiency mill", could the authors explain what these supply chain configurations are and why they have this effect? The sentence follows from a seemingly unrelated topic, that producers and traders retain palm oil to sell during the low season. Sales of palm oil and sales of FFB are very different things. Some rewriting here could help to explain the link between retention of palm oil and the type of mill a producer sells their fresh fruit bunches to.

Lines 338-340, 349-351: The findings presented here are particularly novel and important for the wider field:

- 1) 'Our results suggest that current palm oil sustainability approaches (e.g., RSPO) that target company mills and their surrounding supply shed as a certifiable unit will fall short'
- 2) 'widespread use of mills with low capacities and extraction rates presents a considerable opportunity to increase palm oil yields at the processing stage and achieve major production gains from the existing cultivated area.'

Line 381: How are 'immature oil palm monoculture systems' defined? At what year from planting is a field no longer immature?

Lines 458-460: The methods around the second analysis were not entirely clear to me. The authors state: 'The second [analysis] estimated deforestation due to expansion, relative to the conversion of other land cover types, on a reduced dataset that included only grid cells where expansion occurred (n = 12,266).' Elsewhere, a far larger number of grid cells was mentioned in relation to the conversion to immature monocultures (ten percent being n = 23,347 pixels). It would be good to clarify how there can be fewer grid cells where expansion occurred than the mentioned just above where expansion to immature monoculture took place.

Figure S1: Please provide untransformed units in the axis labels.

I hope these comments are useful to the authors in revising this interesting manuscript.

REFERENCE

Ajambang, W., & Ijang, T. P. (2016) Fruit Crop theft and its impacts on the productivity of oil palm agro-industries in Cameroon. *Developing Country Studies*, 6-5.

Reviewer #3 (Remarks to the Author):

This paper provides a fairly straightforward analysis of the relation between deforestation caused by expansion in oil palm and the number, distribution and properties of milling capabilities in Cameroon, arriving at the conclusions that (1) for a variety of reasons a large part of the

deforestation is associated with small producers and inefficient mills: (2) from the point of view of the local actors there is little reason for this to change and one can expect oil palm expansion by small producers to continue to be a major contributor to deforestation. The description of some of the methods is vague and, as noted below, the mapping estimates are not area-corrected. Though the paper is interesting, the study is for one part of a single country in W. Africa with its own special conditions, and whether the conclusions have any wider generality is not discussed. This wider context is needed.

The paper is quite repetitive and could be readily shortened by 30% without losing significant content.

Detailed comments

Fig. 1 is not a plot of trends, as it says in the caption, but of values of area and yield. Also, from this Fig., the acceleration is around 2005, not 2000 as it says in l. 158. What is the reason for the sharp dip in yield around 2008?

l. 172. Table S2 does not appear to say anything about land allocation.

Not at all clear in the text on p.11 et seq. is the role of the zone of influence in defining the quoted numbers. This concept is only introduced on p.23 but it is essential to understand what the numbers given signify. As a related issue, what does the “predicted probability of deforestation” actually mean (l. 212-213)?

Figure 3. The location of industrial mills is not clear. Please indicate in a) the location of b).

Table 1 effectively states a set of hypotheses (the expected signs) but there are several discrepancies between these signs and what is observed, both as regards oil palm expansion and deforestation due to oil palm expansion, most of which are not discussed.

l. 221-3 seems to contradict the caption of Fig.5, which says that these odds lack significance. The shaded area on Fig 5 is not defined. When an extraction rate is given as a percentage, it is a percent of what?

What does l. 253 actually mean?

l. 264 – 266. No hypothesis is stated (anywhere), and what is actually meant by “consistency with ... phenomena”?

316. What does “due to supply chain configurations” mean?

346. Little if anything is said about S-E Asia and why this should be so different from Cameroon, or whether Cameroon is a special case for which conclusions cannot be readily extended to other parts of Africa.

384. The process for estimating oil palm in immature monoculture is not made clear. Indeed, from the text it is not clear why this immature class can be regarded as monoculture rather than a patchwork of different cover types. Is the estimation of the oil palm part geographically specific or simply a proportion that is not georeferenced?

391 et seq. “Areas of change were identified independently of the classification maps to avoid error propagation (see Supplementary Material)”. This description is missing and also any validation of the change maps. The description of “validation” in the second part of this paragraph is neither clear nor convincing. It appears to say that 23,347 pixels were individually examined visually to check which were immature oil palm (this is certainly not validation). Is this what was really done and how reliable is this process?

461: Weighted how?

General remark re mapping: No area adjusted accuracies have been reported when presenting the mapping results. Overall and class-specific accuracy have been derived straight from the error matrix of sample counts without correcting for the area proportion of the different classes. As the sample size for the different classes are not proportional to the class area, the inclusion probability was therefore not considered. Without reporting area adjusted accuracies the results are statistically not sound and cannot be compared with results from other studies. The authors would be wise to calculate area adjusted accuracy measures following the step by step guidelines presented in: Olofsson, P., Foody, G. M., Herold, M., Stehman, S. V., Woodcock, C.E., & Wulder, M. A. (2014). Good practices for estimating area and assessing accuracy of land change. *Remote Sensing of Environment*, 148, 42-57. <http://doi.org/10.1016/j.rse.2014.02.015>. Furthermore, the same guidelines should be used to estimate the area of each class.

General remarks:

The use of the word “proximity” rather than “distance” is not helpful as it is often not clear what is meant, e.g. does an increase in proximity mean that it is nearer or further? Even less helpful is then to express proximity in negative units as in Fig. 5.

Although odds and probability are taken to mean the same thing, it would be better to use just one of these terms consistently throughout.

Supplementary material

12-18. There appears to be no connection between the numbers in this para. and Table S1.

18. As noted above, this process of verification is not explained properly, and it is not clear if it is a georeferenced process or instead an estimate of proportion.

Figure S4. The implication of these ROC curves is that high rates of detection (say 80%) lead to significant false positive rates. In the landscape, there will be many more pixels that are not converted than are, so the false positives will far outweigh the true detections. How is this dealt with?

Manuscript NCOMMS-18-15572A

RESPONSE TO REVIEWER COMMENTS (Author responses in blue):

Dear Reviewers,

Thank you for your detailed feedback. We have addressed each comment individually. Changes made in the text, figure and table captions, and Supplementary Material are highlighted in yellow. Our responses to each of your comments below are in blue.

Our revisions include clarifying several points in the Methods section, providing greater interpretation of results, and a discussion of their generalizability beyond Southwest Cameroon.

We appreciate the opportunity to revise our manuscript and feel that this version is greatly improved as a result of your comments and suggestions.

Sincerely,

Elsa M. Ordway and coauthors

Reviewer #1 (Remarks to the Author):

The paper entitled 'Oil palm expansion at the expense of forests in Southwest Cameroon associated with proliferation of informal mills' examines the role of informal artisanal mills in oil palm expansion and associated deforestation in Southwest Cameroon. The main claims are:

1. Oil palm expansion in Southwest Cameroon occurs in relation with small-scale informal mills.
2. 2/3 of expansion and deforestation associated to oil palm plantations between 2000 and 2015 was carried out by non-industrial growers.
3. Deforestation is happening, even in this densely populated area of Cameroon.

Studies on oil palm artisanal mills have so far been focusing on technical and economic aspects, but not on land-use change dynamics. Reference is made to most of previous publications on this subject in Cameroon. This study thus addresses artisanal milling from a new and interesting angle.

The method is well described and adapted to the objective of the research. The statistical analysis is appropriate. Results are clearly exposed.

The discussion could benefit from more references and comparison to other studies. In the introduction, previous literature on industrial agriculture in Central Africa and its impact on forests could be deepened.

Land use change analysis focusing on oil palm expansion and associated deforestation at the national and sub-regional scale would be an interesting follow-up to this study. This could be crossed to industrial concession locations to get an idea of the genericity of the results. Of course a repetition of the whole method including field work to characterize artisanal milling would be great, but would require major means and time.

Manuscript NCOMMS-18-15572A

I propose below some comments and suggestions on each part of the paper in the hope that they might help strengthening the research.

Introduction:

- References 16 to 20 (quoted L 73-74): the reader would expect references on African cases in this paragraph, on the five references quoted here only one refers to Africa, and it is an NGO report not a scientific publication.
 - Suggested references: Deininger et al 2011, Megevand 2013, Feintrenie 2014, Oyono et al 2014, Marquant et al. 2015, Feintrenie et al 2016.
 - Thank you for these useful suggestions. We included Deininger et al 2011, Feintrenie 2014, Marquant et al. 2015, Feintrenie et al 2016, given their relevance to the statements made in L 82-87, and we removed the NGO report. We also included reference to a relevant article published in August: Strona et al., 2018 *PNAS*.
- “we explore whether trends in sub-Saharan Africa conform to the widely held view that large, industrial companies drive most oil palm expansion.” (L 76) To explore this hypothesis, a study at a national or sub-regional scale would be useful. At the study site scale, the age of industrial concessions and the level of land saturation and availability to conversion to agriculture (not under a conservation status or allocated to other purpose such as mining or urban development) are decisive to current dynamics of plantation expansion.
 - Although a continental, or even national, scale analysis would be ideal, we chose to focus on a top oil palm producing region of Cameroon to carry out a detailed remote sensing and change detection analysis, combined with on-the-ground mapping of processing facilities. We have revised our description of the study on L 54-56 to clarify the scale and objective of this study. We believe that this is still a relevant scale for analysis, especially given the level of detailed spatial analysis we are able to conduct with data collected on milling facilities. Additionally, we hope that results from this study motivate further research at larger spatial scales and additional sites.
- L84: Is it where the introduction finishes and the ‘Results’ begin? There is a lack of clarity in the plan of the paper.
 - Given the short format of Nature Communications papers, we were somewhat restricted in our ability to follow the more traditional manuscript format. However, we incorporated more signposting at the beginning of the paper to provide more structure for the reader.
 - This included moving the description of the goal of the study to second paragraph (L 54-63) and included a sentence outlining the remainder of the Introduction section (L 61-63). We also clarified where our results begin by including a “Results” section header and starting the sentence on L 186 with “Our results show...”

Results:

- The Southwest region “is 86% forested” (L 162), how much of this is suitable to oil palm? What definition of ‘forest’ is used here?

Manuscript NCOMMS-18-15572A

- Forest was defined as tree cover greater than 50% over an area of at least 0.09 ha (30 m) and included primary and secondary forests. We clarified that the Southwest region was 86% forested in 2000, at the start of the study period. Using the IIASA-GAEZ gridded suitability data on rain-fed oil palm, for a baseline period of 1961-1990, we calculated that 81% of the Southwest region is suitable for oil palm. This information is now included on L 166, and L 429-430.
- A short historical background on the ‘two agro-industrial companies’ (L 164) would be important to understand their current dynamic of expansion. The map covers three companies, two with a long history in the region, and one quite new.
 - We mentioned the more recent presence of Herakles/SGSOC, the third agro-industrial company, in SW Cameroon, and clarified that it is the two more established companies, CDC and Pamol, who own and operate the 5 industrial mills in the region – see L137-139 and L 168-170.
- L 165 ‘companies’ is used for mills, whereas L 164 it is used for industrial firm. The use of this word needs to be consistent.
 - When referring to mills, we revised descriptions throughout the text to distinguish between mill ‘types’ based on efficiency, where efficiency refers to a mills capacity and extraction rate (see Fig. S1). In the manuscript, we now refer to mills as ‘high-efficiency agro-industrial’ mills and ‘low-efficiency informal’ mills. At times we refer to company-owned high- efficiency mills to emphasize that the high- efficiency mills in our study area are all owned by agro-industrial companies (see Abstract, and L 265, 328). However, we removed all instances where mills are only referred to as ‘company mills’ to prevent confusion.
- “expansion occurred outside industrial plantations” (L 181), ‘concessions’ would be more appropriate than ‘plantations’ here, same on L 203, 205, 216, 220, 222, 233 and 237 (Figure 5 title).
 - Where ‘plantation’ refers to agro-industrial plantation, we replaced ‘plantation’ with ‘concession’, and described fully as ‘agro-industrial concession’ the first time the term was mentioned (e.g., L 190, 191, 217, 238, 247, 262, 304, 314, Fig. 3 caption and plots, Fig. 4 plot, Fig. 5 caption and plots, Table 1, Table S2, Table S3)
 - Where ‘plantation’ refers to smallholder/non-industrial plantations, we used ‘non-industrial plantation’.
 - When generally referring to oil palm farms/cultivation, ‘plantation’ was used (L 256)
- Figure 3: Hansen et al (2013), used for the tree cover data in figure 3 (L 192), does not make distinction between natural forest and plantations (including rubber or oil palm plantations). Are land use transitions shown on the figure conversion from natural forest or annual crops/savannas to oil palm, or conversion from natural forests and tree plantations or annual crops/savannas? Is land use change in the CDC concession in Bamuso deforestation of natural forest or conversion from rubber plantations? In the method section (L 366-401), land cover analysis is detailed and does not include the support of Hansen et al (2013) classification. Could you clarify the use of Hansen et al. (2013) tree cover data in figure 3?
 - The Hansen et al (2013) tree cover data were not included in any of the analyses in this study. The tree cover data were displayed in Fig 3 to provide a visual of the gradient in tree cover across the study area, however this has been removed from Fig 3 to prevent

Manuscript NCOMMS-18-15572A

confusion. In this study, forest cover excludes tree plantations, which were classified separately as monoculture agricultural systems – see L 421-432.

- How do you consider land availability in the analysis? Isn't it an explicative variable of oil palm expansion or deforestation? Land availability is related to population density, but not only. Did you exclude from the land-use analysis land that were available to oil palm conversion because of their status (protected, public reserve for industrial or urban development, etc.) or tenure?
 - We agree that land availability is a critically important determinant of oil palm expansion and deforestation. Given the lack of data on land tenure, we were unable to include information on that in this analysis. We did not exclude protected (e.g., National Parks or reserves) or other zoned land-use areas (e.g., community or council forests) because we were interested in quantifying *all* oil palm expansion and associated deforestation across the landscape. In the statistical analyses, we included population density as a variable to quantify the relative influence of mill type and proximity to agro-industrial concessions on expansion, while controlling for the influence of population pressure.

Discussion:

- L 245-252 are repetitive of introduction and results.
 - We feel that, after a lengthy introduction and the reporting of detailed results, it can be helpful to the reader to synthesize key findings in the context of the aim of our study, prior to discussing these findings at greater length. For this reason, we have left L 256-267 in.
- History is referred to but not enough used as explanatory factor of the present location of oil palm plantations and expansion:
 - “We expected to find that economies of scale would lead to greater expansion of large-scale plantations” (L 259). Quite a strange hypothesis knowing the history and age of most of the industrial concessions in the study site, with the exception of SG-SOC concessions.
 - Given the broad readership of Nature Communications, we feel that it is important to explain that, despite the widely reported concern of industrial-scale expansion (e.g., Deininger 2011, Linder 2013, Strona et al. 2018), our findings indicate that expansion in Cameroon is not currently being driven primarily by large, agro-industrial companies, which we feel holds significant policy implications. We have rephrased L 269-272 to frame our findings as counter to the more widely publicized narrative of corporate-driven oil palm expansion. Additionally, given rising palm oil demands within Cameroon, the two well-established agro-industries in the country continue to actively clear new land for oil palm cultivation. Prior to carrying out this study the relative contribution of industrial and non-industrial producers to recent expansion had not been quantified, nor was it entirely clear that smallholder oil palm producers were contributing substantially to deforestation.
 - “concentration of non-industrial expansion near informal milling facilities rather than near high-efficiency mills.” (L263) Why is this surprising in the Cameroonian context? Informal mills develop along the roads, nearby or within villages, and where oil palm small scale plantations are already developed at short distance. Their presence might foster the expansion of plantations if there is enough land available. Is there enough land available to expand oil palm plantations around industrial mills and plantations?

- Even though we were aware of the widespread use of low-efficiency mills in Cameroon, we anticipated greater expansion near higher-efficiency mills given the greater capacity of these mills, which is particularly relevant during peak production season. Based on time spent in the field, talking to farmers and visiting oil palm plantations across the region, we did not see or hear about a shortage of land. Survey work conducted with non-industrial producers across the region (see Ordway et al. 2017, *Global Environmental Change*) indicate that only 2% (14/545) producers who were interviewed reported land availability/scarcity as a barrier. These 14 producers who did report issues of land availability were scattered across the region, including near Mamfe and Kumba, suggesting that there is little relationship between the location of industrial mills and concessions and land availability. Without more directed research on the subject, however, we hesitate to speculate in this manuscript about the influence of land availability on expansion patterns.
- We do agree with your point that the establishment of informal mills and small plantations can be intricately connected, which we discuss in L 295-301.
- “informal mill operations aggregate near company plantations” (L 282), no explanation regarding thefts of FFB from industrial plantations? This is a common complaint of industrial plantations managers in the area...
 - We now mention theft on L 306-309.
- “The establishment of non-industrial producers near plantation estates also relates to direct contractual relationships with companies” (L 301). Are new contracts signed between companies and smallholders in recent days in the study site? Can this factor explain present expansion of small-scale plantations?
 - We do know that contractual relationships between agro-industrial companies and non-industrial producers is actively occurring in different forms, through land-leasing relationships and/or milling and transportation relationships. However, we do not have data on how prevalent this is, and we feel that further research is required to examine this question in more detail. For this reason, we only briefly mention it as a *possible* explanation.
- L310 and following: it would be good to link rainy seasonality to seasonality of prices for the non-informed readers.
 - We have attempted to clarify this link between precipitation seasonality and price seasonality in L338-340.
- “Our results suggest that current palm oil sustainability approaches (e.g., RSPO) that target company mills and their surrounding supply shed as a certifiable unit will fall short in this context.” (L339) Which result does suggest this? No reference to support the statement?
 - We clarified L 368-370 to highlight the results that lead to this conclusion.
 - Also, see Fig. 3 for the restricted locations of industrial mills relative expansion that occurs throughout the region, and results described in L 212-214, L 229-233 and L 264-265.
- It is sad that the discussion doesn’t give any clue to the genericity of the results. To Cameroon? To Central and West Africa? To non-african producing countries? The last paragraph of Discussion quotes Southeast Asia and sub-saharan Africa, but no literature is quoted to discuss a comparison.

Manuscript NCOMMS-18-15572A

- We have now included some discussion of the broader relevance of our findings in Cameroon, in comparison with other oil palm producing countries in Africa, and relative to Southeast Asia. See L 370-394.

Laurène Feintrenie
Forests and Societies research unit
CIRAD, Univ Montpellier, France
Associate scientist to CATIE and to ICRAF
CGIAR Research Program on Forests, Trees and Agroforestry
CATIE, Turrialba, Costa Rica.

Reviewer #2 (Remarks to the Author):

Summary

Destruction of rain forests for the production of palm oil has been a strong driver of biodiversity loss. While most of the oil palm-related deforestation since 2000 has taken place in Southeast Asia, there is increasing concern about sustainability issues around new oil palm frontiers in Africa and South America. This manuscript by Ordway et al. presents new and relevant insights into oil palm expansion dynamics and related deforestation in a part of Sub-Saharan Africa. The authors have conducted a spatially explicit analysis of oil palm expansion between 2000-2015 in the Southwest province of Cameroon. Using Landsat satellite imagery, they measured the contribution of oil palm expansion to deforestation in and around industrial concessions. Furthermore, they analysed the relationship between different milling systems and expansion using the first georeferenced dataset of palm oil processing facilities for the region.

Using this case study of a small palm oil producing region in Cameroon, they argue that non-industrial palm oil producers – smallholder farmers and small-scale palm oil millers – are key players in palm oil production and associated deforestation. Two thirds of expansion and deforestation was due to non-industrial oil palm farms. Not only do these findings shed light on key differences in the dynamics of land use change relative to the Southeast Asian experience, it also shows that this type of small-scale production deserves more attention from civil society and researchers working on deforestation issues. Another important contribution of the manuscript is the finding that new oil palm plantations in the region are predominantly replacing forests (67%), something which has not been shown before.

The emphasis on the presence of milling facilities of the manuscript is a welcome addition to the literature. Milling is a critical step in palm oil production and considering oil palm development pathways should include consideration of this factor.

A particular strength of the study is the georeferenced dataset of all the 498 palm oil mills in the region – ranging from tiny mills with a manual press, to factory-like mills belonging to industrial plantation companies – which has been collected during multiple field visits and contained data on various aspects of the mills.

This manuscript argues rightly that the widely publicized narrative of industrial-scale oil palm expansion should be revised: at least in this region, most oil palm expansion and associated

Manuscript NCOMMS-18-15572A

deforestation occurred outside large plantations. Hence, there needs to be increased attention to the potentially harmful effects of NGO and government schemes promoting smallholder oil palm farming; ironically, these are sometimes touted as a tool to improve sustainability. In addition, the manuscript points out that major sustainability gains could be made by improving the extraction rate of the small informal mills.

Overall, I believe that the manuscript is a convincing study of a previously neglected aspect of oil palm development. It is well structured and clearly written. The work has a number of implications for our understanding of deforestation by non-industrial palm oil producers; this is relevant not only to other researchers but also to governments and civil society.

I do, however, have some concerns about the presentation and interpretation of results, see details below.

MAJOR POINTS

1. How confident are the authors that the large amount of aggregated forest-to-oil-palm conversion in the top left corner of figure 3b is not actually part of a recent expansion by Pamol (a company)? The pattern looks much more like an industrial plantation than smallholder expansion.

The WRI Forest Atlas which was used to delineate plantation areas is a citation from 2012. On the interactive atlas online it was not easy to find details on whether this map is regularly updated and if not, what year the plantation boundaries were measured. If the map was updated in, say, 2010, this study misses five years of company expansion. Expansion certainly took place in that period, although I do not have exact information on its location. The nearest company mill is theoretically near enough to the area mentioned above for it to be used for a new plantation. It would be good for the authors to clarify this and present some evidence that this area is indeed smallholder expansion and not a new industrial plantation.

- We appreciate you raising this concern, as we had similar thoughts while conducting this study. To the best of our knowledge, the large area of expansion in Fig. 3b is not recent expansion by Pamol. We have reached out to staff at the Ministry of Agriculture (MINADER) and Ministry of Forests and Wildlife (MINFOF), as well as WRI staff to inquire about this. While we were told about a 1000 ha Pamol “extension” area, we have been informed by multiple, independent sources that this “extension” is planned for the Bakassi Peninsula, and is in the planning stages. Our sources told us that no land clearing has been carried out yet, which is consistent with our finding of no oil palm expansion on the Bakassi Peninsula from 2000-2015. One source mentioned a small CDC “extension” south of Korup NP, although this area also does not appear to have been cleared/converted to date. There remains the possibility that the aggregated area of expansion in Fig. 3b could be mixture of smallholder plantations and independent producers contracted by companies. However, further research investigating those relationships in more detail is required to address that question, which we feel is beyond the scope of this study. We are also aware of CDC plans for extension in Boa Plain. However, that is being carried out near Iloani, much further south than area of expansion in the top left of Fig. 3b.
- Regarding the quality of the WRI plantation boundary data, we acquired the most up-to-date version of plantation boundaries from WRI staff in DC and Cameroon in 2015. We followed up with them while conducting these revisions to verify that they have not since updated the dataset. They have not, and we are therefore using the most current, available information.

Manuscript NCOMMS-18-15572A

2. The method for validating immature oil palm monoculture pixels is not very clear and should be rewritten for better understanding. The authors state: ‘Ten percent of all pixels characterized as conversion from forest or other land cover type in 2000 to immature monoculture in 2015 were randomly selected for validation (n = 23,347).’ Considering that it appears that the increase in oil palm from 2000 to 2015 was mostly attributable to immature and not mature plantation (see supplementary material), the issue of how exactly the immature plantation was classified and mapped is very important. Therefore, please include in the methods:

a. a description of the method of classification and mapping for the remaining 90% of immature monoculture pixels

- We have added a much lengthier description of the estimation of the fraction of immature monoculture that was immature oil palm in 2015. Our previous use of the term “validation” was incorrect, and we have revised this description to clarify what we did. The immature monoculture class was not re-mapped. Given our inability to spectrally distinguish immature oil palm as a class, our goal was to estimate the fraction of areas identified as transitioning to immature monoculture in 2015 that were specifically a transition to immature oil palm (as opposed to immature banana or rubber). This procedure is now described in more detail in L 457-477.

b. information on what proportion of the oil palm in 2015 was immature vs mature

- Based on this estimated fraction of immature oil palm within the immature monoculture class, we calculated that 45.26% of land area converted to oil palm in 2015 was mature and 54.74% was immature. This is now included in L 475-477.

c. information on what proportion of the area which was forested in 2000 but immature monoculture in 2015 was immature oil palm versus immature other crops

- Based on the estimated fraction, 92.78% of the area forested in 2000 that transitioned to immature monoculture systems, was a transition to immature oil palm, while 7.22% was a transition from forest to some other type of immature monoculture system. This is now included in L 473-475.

3. I had some difficulty understanding some of the results, and wonder if the Y axis labels in Figure 5 are mis-labelled. As I understand from the methods, these plots show the odds ratio – the probability of deforestation for oil palm, relative to the probability of deforestation for other land covers. A probability of 1 for deforestation far from informal mills, would therefore mean that deforestation for oil palm is as likely as that for other land covers. If this interpretation is correct, the axis titles should be edited, and whether or not it is correct, the meaning and calculation of odds ratios in this paper needs to be better explained. If interpreted as a probability of 1, as the axis title indicates, it would imply that all forest pixels were deforested, even tens of km from informal mills, which is not credible. Please check and relabel as appropriate.

- Fig. 5 shows predicted likelihoods, not the odds ratios (which are shown in Fig. 4). The predicted likelihoods in Fig. 5 are plotted using partial dependence plots (Friedman 2001, *Annals of Statistics*) that illustrate, in this case, how the likelihood of oil palm expansion (a, c) and deforestation resulting from expansion (b, d) vary as a function of distance to informal mill (a, b) and mill extraction rate (c, d), holding all other variables constant at their mean. Thus, these are the predicted mean likelihood responses *on average*, and should be carefully interpreted as such. Additionally, we now show the frequency distribution of the variable on the x-axis of each panel in Figure 5 to help with interpretation. The frequency of data for a given variable in the model are not high where the data for that variable are scarce, and are non-

existent beyond these distributions, so it is important not to place too much on interpretation where the predictions are less reliable (i.e., where there is limited or no data).

- We have provided additional interpretation of odds ratios and likelihoods in the text in L 205-209 and the caption for Fig. 4, and we revised the use of language in the text to clarify the different interpretation of predicted mean likelihoods from odds ratios. For example, see L 212-216. We also removed Fig. 5 from L 204-205 to clarify that Fig. 5 does not refer to the odds ratios, and we interpreted those figures in the Results separately.
- Additionally, we replaced the word “probability” in L 265 and removed it entirely from L 550 in the Methods section to avoid confusion.

4. The authors’ dataset indicates that informal mills are mostly very small in capacity (Figure S1). Given this, and the fact that FFB can be supplied to industrial mills from tens of km away, it is still plausible that deforestation for oil palm a short distance from plantations is partly to supply industrial mills. Lack of deforestation in the immediate vicinity of these industrial mills could be explained by past land use conversion, limiting the options for new oil palm. More caution in attributing land-use change to informal mills may therefore be warranted. In particular, a comparison of the total capacity of the five industrial mills, and how that has changed through time, with the total capacity of the informal mills, would be useful.

- Reviewer 1 raised a similar question of land availability near industrial mills. Our response to Reviewer 1 is included the following:
 - Based on time spent in the field, talking to farmers and visiting oil palm plantations across the region, we did not see or hear about a shortage of land. Survey work conducted with non-industrial producers across the region (see Ordway et al. 2017, *Global Environmental Change*) indicate that only 2% (14/545) producers who were interviewed reported land availability/scarcity as a production/expansion barrier. These 14 producers who did report issues of land availability were scattered across the region, including near Mamfe and Kumba, suggesting that there is little relationship between the location of industrial mills and concessions and land availability. Without more directed research on the subject, however, we hesitate to speculate in this manuscript about the influence of land availability on expansion patterns.
- Additionally, we would like to point out that there is still active conversion both inside and outside concessions near industrial mills. However, we see more conversion outside concession and there is a stronger relationship between conversion and informal mills, suggesting a greater relative role of these mills. This is also illustrated by the observed expansion throughout the region, in close correlation with informal mills, while industrial mills are restricted to the southern third of the region.
- We agree that producers will travel tens of kilometers to supply their FFBs to a mill, and we extrapolate on the possibility that some fraction of producers likely supply to both industrial and informal mills in L 349-351.
- We also included a comparison of the total capacity of industrial and informal mills and report the contribution of informal mills to increased capacity since 2000 in L 194-200.

5. While the authors have indicated that mill data will be made available on request, in this day and age it is reasonable to expect such data to be made available in a more permanent and organised way through an online repository. If there are ethical concerns about making the mill data publicly available, this may not be appropriate, but if not, it would be good to see the data provided through a repository.

Manuscript NCOMMS-18-15572A

- We are unfortunately unable to make the mill dataset publicly available. In the process of looking into making these data public, we ultimately decided that even upon removing personal identifying information, the mill GPS points provide identifying information that could be used against respondents. To protect the identity of respondents who provided us with this information and to honor our confidentiality agreement with them, we can only provide the data upon request at an aggregated scale (the Division municipality scale). We have updated the Data Availability statement to clarify this. In sum, we can make the original dataset available for review purposes only.

MINOR POINTS

- Line 66 – 72: This information on current sustainability efforts feels tangential rather than essential.
 - We feel that this research provides new insights into the role that non-industrial producers (i.e. non-corporate actors) play in oil palm expansion in SW Cameroon. This finding has important policy implications, which we discuss in L 365-384 in the Discussion section. Given the broad readership of Nature Communications, we have included L 80-88 in the Introduction to orient palm oil sustainability issues and efforts in Africa relative to broader, international efforts to improve the sustainability of the sector.
- Line 90 – 94: The manuscript sets up its hypothesis in the context of the theory of ‘economies of scale’. While the findings are presented as surprising in relation to this hypothesis, the discussion mentions some very clear reasons why in the Cameroonian case study the economy of scale argument was unlikely to be valid. Therefore, it may not be necessary to include this economies of scales argument and it could be removed for the sake of brevity.
 - Given the broad, interdisciplinary readership of Nature Communications, we feel that it is important to report the theoretical underpinnings of this research. In doing so, we feel that we can more effectively demonstrate concrete ways in which the narrative in Cameroon is more nuanced than the often cited concern of industrial-scale expansion (e.g., Deininger 2011, Linder 2013, Strona et al. 2018). We present our results on the role of non-industrial producers in the agricultural sector and the importance of informal economies in terms of how they do or do not fit the theoretical concept of scale economies, illustrating the applicability and shortcomings of the theoretical framework in the context of Cameroon. Given that non-industrial producers and informal economies play large and important roles in the agricultural sector in other African countries, we feel that this is a useful perspective for both researchers and practitioners.
 - We have rephrased L 97 to in an effort to clarify the theoretical influence of scale economies on producer decision making broadly, allowing us to interpret and discuss results in a manner that is more generalizable.
- Line 151: Figure 1, it would be interesting to be able to see how this local yield relates to yields and yield improvements over time across the planet.
 - We included a dotted line representing the FFB yield in Malaysia, a top producing country with the highest yield.
- Line 162: The study encompasses 40% of national palm oil production. While the region chosen is a key area for both industrial and smallholder oil palm, the Littoral region is of great importance

too. Cameroon's only privately owned large industrial company group is located in the Littoral (the two plantation companies mentioned in this study are part state-owned). This company is run rather differently than the two in the Southwest and so one could hypothesise that the relationships with smallholder farmers would be very different. If true, this could affect the expansion dynamics around the larger company milling facilities and thus the conclusions of this study. Perhaps something along these lines should be noted in the discussion. At minimum, the methods should state why the Southwest was chosen over other regions.

- We have now included a statement in the Methods section about why we excluded the Littoral and Central regions from this analysis. See L 409-411. We also briefly discuss the relevance of our findings to the rest of Cameroon and other oil palm producing countries in Africa in L 386-394.
- Line 190: Figure 3:
In the legend, where it states plantation, presumably industrial palm oil plantation is meant. This should be clarified. Are these only oil palm plantations or is e.g. rubber, banana etc also included? Why are smaller areas planted with oil palm not also called a plantation? What is the size limit above which it is called plantation, or what other definition do the authors use? In Cameroon, oil palm smallholdings entities can be up to 500 hectares large (often owned by rich city dwellers, run like a mini industrial plantation).
 - We have clarified this terminology in Fig. 3 and throughout the text, so that agro-industrial oil palm plantations are now specifically referred to as “concessions”. Plantation can refer to an oil palm farm of any size. We also included a statement in the Methods section on L 520-522 explaining that these boundaries are inclusive of all concession areas leased by agro-industries and can be used for cultivating oil palm, rubber, or banana.
- Line 281 – 286: A factor which should be included here in relation to the potential reasons for informal mills agglomerating around newly developed parts of industrial plantations is the widespread practice of stealing fresh fruit bunches (FFBs) from company land (see for example Ajambang and Ijang 2016). The presence of informal mills close to newly developed oil palm within company plantations rather than older areas within the plantations may well have to do with the fact that it is easier to steal FFBs from younger palms because they are shorter and are typically located further from the main company infrastructure.
 - REFERENCE: Ajambang, W., & Ijang, T. P. (2016) *Fruit Crop theft and its impacts on the productivity of oil palm agro-industries in Cameroon. Developing Country Studies*, 6-5.
 - Thank you for bringing this citation to our attention. We now reference this article and mention FFB theft in the context of informal mill proximity to industrial concessions actively expanding their oil palm area in L 306-309.
- Lines 316-318: Description of the issues surrounding seasonal changes in price and production is not very clear. Rather than write “Due to supply chain configurations, supplemental income from this type of storage is not possible when a producer sells their FFB to a high-efficiency mill”, could the authors explain what these supply chain configurations are and why they have this effect? The sentence follows from a seemingly unrelated topic, that producers and traders retain palm oil to sell during the low season. Sales of palm oil and sales of FFB are very different things. Some rewriting here could help to explain the link between retention of palm oil and the type of mill a producer sells their fresh fruit bunches to.
 - We included additional explanation in L 338-340 and L 342.

Manuscript NCOMMS-18-15572A

- Lines 338-340, 349-351: The findings presented here are particularly novel and important for the wider field:
 - 1) ‘Our results suggest that current palm oil sustainability approaches (e.g., RSPO) that target company mills and their surrounding supply shed as a certifiable unit will fall short’
 - 2) ‘widespread use of mills with low capacities and extraction rates presents a considerable opportunity to increase palm oil yields at the processing stage and achieve major production gains from the existing cultivated area.’
- Line 381: How are ‘immature oil palm monoculture systems’ defined? At what year from planting is a field no longer immature?
 - Mature oil palm systems refer to oil palm plantations that are roughly 4 years or older, after which point palm fronds have developed and the canopy is closed. Thus, immature oil palm included in the immature monoculture class refers to plantations within the first three years after planting, when palm fronds are still developing, and the canopy is not yet closed, leading to spectral confusion with other immature monoculture classes given the relatively large proportion of bare soil being reflected.
 - This information is now included in L 438-445.
- Lines 458-460: The methods around the second analysis were not entirely clear to me. The authors state: ‘The second [analysis] estimated deforestation due to expansion, relative to the conversion of other land cover types, on a reduced dataset that included only grid cells where expansion occurred (n = 12,266).’ Elsewhere, a far larger number of grid cells was mentioned in relation to the conversion to immature monocultures (ten percent being n = 23,347 pixels). It would be good to clarify how there can be fewer grid cells where expansion occurred than the mentioned just above where expansion to immature monoculture took place.
 - Transitions based on the original land use/land cover (LULC) classification maps, pre-change detection, were used to estimate the fraction of pixels within the immature monoculture class that were immature oil palm (n = 23,347). See L 457-477 or a more thorough description of this process.
 - The change detection analysis was involved identifying biophysical areas of change using vegetation indices and image differencing. This resulted in a dataset of “change” pixels that was less than the number of pixels identified as undergoing a transition based on a comparison of the LULC maps alone. Both logistic regressions used this change detection dataset, however, the oil palm expansion analysis included all grid cells, so while the number of oil palm expansion grid cells was reduced based on the change detection analysis, the total number of grid cells (including no-change) remained was the same. The second logistic regression analysis, modeling deforestation, was carried out only using oil palm expansion grid cells based on the image differencing change detection analysis. Between the reduced set of change pixels and excluding no-change pixels, this dataset was substantially smaller.
 - We have included more detail in the Methods section to clarify this. See L 479-493.
- Figure S1: Please provide untransformed units in the axis labels.
 - Revised: Fig. S1a now shows the untransformed mill capacity frequency distribution.

I hope these comments are useful to the authors in revising this interesting manuscript.

Reviewer #3 (Remarks to the Author):

This paper provides a fairly straightforward analysis of the relation between deforestation caused by expansion in oil palm and the number, distribution and properties of milling capabilities in Cameroon, arriving at the conclusions that (1) for a variety of reasons a large part of the deforestation is associated with small producers and inefficient mills: (2) from the point of view of the local actors there is little reason for this to change and one can expect oil palm expansion by small producers to continue to be a major contributor to deforestation. The description of some of the methods is vague and, as noted below, the mapping [error??] estimates are not area-corrected. Though the paper is interesting, the study is for one part of a single country in W. Africa with its own special conditions, and whether the conclusions have any wider generality is not discussed. This wider context is needed.

The paper is quite repetitive and could be readily shortened by 30% without losing significant content.

Detailed comments

- Fig. 1 is not a plot of trends, as it says in the caption, but of values of area and yield. Also, from this Fig., the acceleration is around 2005, not 2000 as it says in l. 158. What is the reason for the sharp dip in yield around 2008?
 - We have removed the use of the word trend in the Fig. 1 caption, and rephrased L 161 to specify that acceleration began around 2005.
 - We had the same question about the dip in yield around 2008 and have not been able to obtain an answer from Cameroon Ministry of Agriculture staff or other knowledgeable experts. Based on feedback from several experts, we believe the most likely explanation is that the dip is an artifact of poor data quality. We have included a more explicit caveat that these yield data should be viewed with caution given the difficulty in obtaining good quality data on FFB yields, especially from smallholder producers. We include them because they are the best available data on temporal changes in yield to provide a comparison against production and area expansion increases. USDA data on oil palm FFB yields in Cameroon are only available after 2000, and although the actual USDA yield values are also of poor quality, they show a similar decline in the late 2000s.
- l. 172. Table S2 does not appear to say anything about land allocation.
 - We have rephrased L 178 to clarify that we are referring to concession areas leased by agro-industries.
- Not at all clear in the text on p.11 et seq. is the role of the zone of influence in defining the quoted numbers. This concept is only introduced on p.23 but it is essential to understand what the numbers given signify. As a related issue, what does the “predicted probability of deforestation” actually mean (l. 212-213)?

Manuscript NCOMMS-18-15572A

- We moved an explanation of the mill extraction rate variable into main text in L 179-181. We also omitted the use of “probability” and clarified that panels in Fig. 5 illustrate partial dependence plots of the predicted mean likelihood of expansion and deforestation as a function of two variables of interest in each model. This is now explained in more detail in the captions of Fig. 4 and 5.
- Figure 3. The location of industrial mills is not clear. Please indicate in a) the location of b).
 - We have revised Fig. 3 to clarify the location of industrial mills and we included a box denoting the location of (b) in panel (a).
- Table 1 effectively states a set of hypotheses (the expected signs) but there are several discrepancies between these signs and what is observed, both as regards oil palm expansion and deforestation due to oil palm expansion, most of which are not discussed.
 - Thank you for bringing this to our attention. These hypotheses were more relevant to the interpretation of our results in an earlier draft of this manuscript. We have removed this column in Table 1.
- l. 221-3 seems to contradict the caption of Fig.5, which says that these odds lack significance. The shaded area on Fig 5 is not defined. When an extraction rate is given as a percentage, it is a percent of what?
 - Panel (a) in Fig. 5, which excludes oil palm expansion *inside* agro-industrial concessions is based on the lack of significance in the first logistic regression, modeling expansion vs. no expansion. Results from the logistic regression modeling deforestation vs. no deforestation due to oil palm expansion did indicate a significant relationship between deforestation *inside* concessions and distance to informal mill, which is why this is included in panel (b). See also Table 1 for results from each model. We revised the wording in L 247-249 in an attempt to make this clearer.
 - We have now clarified in the caption of Fig. 5 that the shaded area is 95% confidence interval around the predicted mean likelihood.
 - Extraction rate is the percentage of the weight of palm oil recovered from a known weight of fresh fruit bunches processed. This is now defined in L 507-510.
- What does l. 253 actually mean?
 - We have expanded on this statement in L 275-284.
- l. 264 – 266. No hypothesis is stated (anywhere), and what is actually meant by “consistency with ... phenomena”?
 - We have rephrased L 285-288 to clarify that these findings are unexpected based on market efficiencies associated with economies of scale that would suggest a stronger relationship between expansion and high-efficiency mills, rather than informal mills.
 - Additionally, we revised the subsequent statement to read, “...consistent with other research demonstrating the important role of informal economies in sub-Saharan Africa”
- l. 316. What does “due to supply chain configurations” mean?

Manuscript NCOMMS-18-15572A

- We have clarified that we are referring to the inability of producers selling to industrial mills to recuperate palm oil that can be sold as a value-added product.
- 346. Little if anything is said about S-E Asia and why this should be so different from Cameroon, or whether Cameroon is a special case for which conclusions cannot be readily extended to other parts of Africa.
 - We have included a discussion of the differences between Southeast Asia and Cameroon, and the relevance of these findings to other oil palm producing countries in Africa. See L 368-380 and L 386-394.
- 384. The process for estimating oil palm in immature monoculture is not made clear. Indeed, from the text it is not clear why this immature class can be regarded as monoculture rather than a patchwork of different cover types. Is the estimation of the oil palm part geographically specific or simply a proportion that is not georeferenced?
 - We have added a much lengthier description of the estimation of the fraction of immature monoculture that was immature oil palm in 2015. Our previous use of the term “validation” was incorrect, and we have revised this description to clarify what we did. Given our inability to spectrally distinguish immature oil palm as a class, our goal was to estimate the fraction of areas identified as transitioning to immature monoculture in 2015 that were specifically a transition to immature oil palm (as opposed to immature banana or rubber). This procedure is now described in more detail in L 457-477.
- 391 et seq. “Areas of change were identified independently of the classification maps to avoid error propagation (see Supplementary Material)”. This description is missing and also any validation of the change maps. The description of “validation” in the second part of this paragraph is neither clear nor convincing. It appears to say that 23,347 pixels were individually examined visually to check which were immature oil palm (this is certainly not validation). Is this what was really done and how reliable is this process?
 - We have now included the previously missing description that describes the change detection analysis in more detail. See L 479-493.
 - Additionally, as described above, our previous use of the term “validation” was incorrect, and we have revised the immature monoculture / immature oil palm description to clarify what we did.
- 461: Weighted how?
 - A weight variable was calculated by summing all no-expansion pixels and all expansion pixels and assigning no-expansion pixels a weight of the sum of expansion pixels divided by the sum of all pixels included in the analysis. Expansion pixels were assigned a weight equivalent to the sum of no-expansion pixels divided by the sum of all pixels in the analysis. The weight variable was included in the model by assigning the calculated weights within the glm function in R.
 - This is now included in the Methods section in L 558-563.
- General remark re mapping: No area adjusted accuracies have been reported when presenting the mapping results. Overall and class-specific accuracy have been derived straight from the error matrix of sample counts without correcting for the area proportion of the different classes. As the sample size for the different classes are not proportional to the class area, the inclusion

probability was therefore not considered. Without reporting area adjusted accuracies the results are statistically not sound and cannot be compared with results from other studies. The authors would be wise to calculate area adjusted accuracy measures following the step by step guidelines presented in: Olofsson, P., Foody, G. M., Herold, M., Stehman, S. V., Woodcock, C.E., & Wulder, M. A. (2014). Good practices for estimating area and assessing accuracy of land change. *Remote Sensing of Environment*, 148, 42-57. <http://doi.org/10.1016/j.rse.2014.02.015>. Furthermore, the same guidelines should be used to estimate the area of each class.

- We appreciate the specific reference to calculating area-weighted accuracies. We have now included the correct error matrix (an older version was previously included that did not correspond to values reported in the text) and recalculated the model and class-level area adjusted accuracies using the method described in Oloffson et al 2014. We also removed the kappa coefficient based on recommendations in Oloffson et al 2014.

General remarks:

- The use of the word “proximity” rather than “distance” is not helpful as it is often not clear what is meant, e.g. does an increase in proximity mean that it is nearer or further? Even less helpful is then to express proximity in negative units as in Fig. 5.
 - We have removed the use of the word “proximity” and now only refer to “distance”. This change is reflected in the text (e.g. see L 212-216), Tables 1, S2, S3, and Fig. 4&5.
- Although odds and probability are taken to mean the same thing, it would be better to use just one of these terms consistently throughout.
 - We now refer exclusively to odds (e.g., see changes made in L 230), except when referring to Fig. 5, which shows the changes in the predicted mean likelihood of oil palm expansion and deforestation as a function of changes in distance to informal mill and mill extraction rate.

Supplementary material

- 1.12-18. There appears to be no connection between the numbers in this para. and Table S1.
 - The numbers previously included were from an earlier model run. We have updated the numbers in the text to reflect the area-weighted accuracies associated with the classification results included in this study, and in Table S1.
- 1.18. As noted above, this process of verification is not explained properly, and it is not clear if it is a georeferenced process or instead an estimate of proportion.
 - It is an estimate of a proportion. See L 457-468.
- Figure S4. The implication of these ROC curves is that high rates of detection (say 80%) lead to significant false positive rates. In the landscape, there will be many more pixels that are not converted than are, so the false positives will far outweigh the true detections. How is this dealt with?
 - The purpose of this analysis is inference. Specifically, we are interested in using these models to examine the relationships between specific covariates and our response variables, and to identify the significance of those covariates, controlling for other

factors. Because the objective of this study is not to make predictions using these models, we feel that AUC values of 0.80 and 0.74 are adequate for the purposes of this analysis. If we aimed to use these models for predictive purposes, we would strive for a higher AUC to improve each model's overall predictive capacity. However, a model with an AUC of 0.7 versus a model with an AUC of 0.9 will not change the significance of the variables of interest. Therefore, we have included the ROC curves and AUC values to demonstrate to the reader that the models do a reasonable job of modeling the outcome/response variable. However, in the Results and Discussion, we focus on the significant covariates of interest (milling variables) and their relationships with oil palm expansion and deforestation.

- A clarification of this objective is now included in L 596-600.

REVIEWERS' COMMENTS:

Reviewer #1 (Remarks to the Author):

The points raised in the previous round of review have been satisfactorily addressed.

Reviewer #2 (Remarks to the Author):

After considering the authors' response to my comments, as well as the changed manuscript, I believe that the paper is ready for publication.

Major changes were made in the description in the methods, which are now far clearer.

In addition, I was particularly interested in finding the increased availability of information in the manuscript which was derived from the survey of the mills e.g. L 193 -199. This makes the study a richer source of information and will help it be of more use to people specifically interested in the state of the Cameroonian oil palm industry. (If there is more such information available I would encourage the authors to publish this elsewhere, if not scientifically, perhaps in a blog or on a researcher's personal website.)

The explanation of the low-efficiency mills are typically manual presses will help readers without experience with these types of mills to picture better the difference between the two types of mills.

I highly recommend the publication of this very interesting piece of research.

Reviewer #3 (Remarks to the Author):

This revised version is much more compact without losing any content, and is much clearer, including in its use of terminology. My concerns about methods and how they were (or were not)

described have been dealt with. Also the description of the overall context of this study has been much improved.

In this form, the paper is much more interesting and readable than the original and its significance is also much clearer.

All the issues I raised have been adequately dealt with and I would recommend publication as is, with just a few tiny points to deal with, as given below.

1. I think the companies should be named at lines 138 and 141, and I think most readers would prefer this.

2. l.169, "company" omitted after industry

l.204: surely "equal to 1" is not correct

272:"their" should be "its"

510. uc T for Thiessen

Supp. Mat.

l.24: averaged across what?

l.26: Full stop is needed before "ROC". I was struggling to make sense of the sentence until I realised it was actually 2 sentences.

REVIEWERS' COMMENTS:

Reviewer #1 (Remarks to the Author):

The points raised in the previous round of review have been satisfactorily addressed.

Reviewer #2 (Remarks to the Author):

After considering the authors' response to my comments, as well as the changed manuscript, I believe that the paper is ready for publication.

Major changes were made in the description in the methods, which are now far clearer.

In addition, I was particularly interested in finding the increased availability of information in the manuscript which was derived from the survey of the mills e.g. L 193 -199. This makes the study a richer source of information and will help it be of more use to people specifically interested in the state of the Cameroonian oil palm industry. (If there is more such information available I would encourage the authors to publish this elsewhere, if not scientifically, perhaps in a blog or on a researcher's personal website.)

The explanation of the low-efficiency mills are typically manual presses will help readers without experience with these types of mills to picture better the difference between the two types of mills.

I highly recommend the publication of this very interesting piece of research.

Reviewer #3 (Remarks to the Author):

This revised version is much more compact without losing any content, and is much clearer, including in its use of terminology. My concerns about methods and how they were (or were not) described have been dealt with. Also the description of the overall context of this study has been much improved.

In this form, the paper is much more interesting and readable than the original and its significance is also much clearer.

All the issues I raised have been adequately dealt with and I would recommend publication as is, with just a few tiny points to deal with, as given below.

1. I think the companies should be named at lines 138 and 141, and I think most readers would prefer this. – We included the companies names at L XXX and L XXX.

2. 1.169, "company" omitted after industry – Revised

1.204: surely "equal to 1" is not correct – Revised to read “do not overlap with 1”

272:"their" should be "its" – Revised

510. uc T for Thiessen – Revised

Supp. Mat.

1.24: averaged across what? – We revised the wording to clarify that we averaged across all 10-fold cross validation hold out sets.

1.26: Full stop is needed before "ROC". I was struggling to make sense of the sentence until I realised it was actually 2 sentences. – Revised